# Model for leisure boat activities and emissions - implementation for the Baltic Sea

Lasse Johansson[1], Erik Ytreberg[4], Jukka-Pekka Jalkanen[1], Erik Fridell[2], K. Martin Eriksson[4], Maria Lagerström[4], Ilja Maljutenko[3], Urmas Raudsepp[3], Vivian Fischer[6] and Eva Roth[5]

[1]Finnish Meteorological Institute, Erik Palmenin aukio 1, 00101 Helsinki, Finland
[2]IVL Swedish Environmental Research Institute, Aschebergsgatan 44, 41133 Gothenburg, Sweden
[3]Marine Systems Institute, Tallinn University of Technology, Ehitajate tee 5, 12616 Tallinn, Estonia
[4]Chalmers University of Technology, Gothenburg, 41296, Sweden
[5]University of Southern Denmark, Esbjerg, Niels Bohrs Vej 9-10, 6700, Denmark
[6]Helmholtz-Zentrum Geesthacht, Max-Planck-Straße 1, Germany

*Correspondence to:* Lasse Johansson (lasse.johansson@fmi.fi)

**Abstract.** The activities and emissions from leisure boats at the Baltic Sea have been modelled in a comprehensive approach for the first time, using a new simulation model leisure boat emissions and activities simulator (BEAM). The model utilizes survey data to characterize the national leisure boat fleets. Leisure boats have been categorized based on their size, usage and engine specifications and for these sub-categories emission factors for $NO_x$, $PM_{2.5}$, CO, NMVOCs and releases of copper (Cu) and zinc (Zn) from anti-fouling paints have been estimated according to literature values. The modelling approach also considers the temporal and spatial distribution of leisure boat activities, which are applied to each simulated leisure boat separately. According to our results the CO and NMVOC emissions from leisure boats, as well as Cu and Zn released from antifouling paints, are significant when compared against the emissions originating from registered commercial shipping in the Baltic Sea. CO emissions equal 70 % of the registered shipping emissions and NMVOC emissions equal 160 % when compared against the modelled results at the Baltic Sea in 2014. Modelled $NO_x$ and $PM_{2.5}$ from the leisure boats are less significant compared to the registered shipping emissions. The emissions from leisure boats are concentrated on the summer months of June, July and August and are released in the vicinity of inhabited coastal areas. Given the large emission estimates for leisure boats, this commonly overlooked source of emissions should be further investigated in greater detail.

Keywords: shipping emissions, leisure boats, anti-fouling paint leach, the Baltic Sea

## 1. Introduction

Shipping activities and emissions for the global commercial fleet can be estimated with modelling approaches that utilize AIS-data and combine this activity data with vessel's technical details (Jalkanen et al. 2012, Johansson et al., 2017). The vessel activities are well known due to the availability and high update rate of AIS-data and these activities can be combined with ship specific technical description. Together, these information sources facilitate the estimation of instantaneous water resistance, engine power use, fuel consumption and ultimately the emissions for each vessel. However, for private leisure boats there are no such direct activity data available that could be used to quantify the emission of air pollutants or water emissions of e.g. toxic compounds, so called biocides, from antifouling paints. Unfortunately, even top-down approaches for leisure boat emission estimation are difficult to utilize since reliable fuel consumption data for leisure boats do not exist. As a consequence, emission inventories with temporal and spatial variability for the leisure boat fleet do not exist.

Since there are several hundred thousand leisure boats being actively used at the Baltic Sea in Sweden alone (Swedish Transport Agency (2010, 2015)) and their activities are mostly situated near populated coastal areas, there is a demand for detailed emission inventories for the leisure boat fleet. Due to its semi-enclosed properties, low biodiversity and slow water exchange, the Baltic Sea is considered to be particularly sensitive to pollution (Tedengren and Kautsky 1987). According to the latest integrated assessment of hazardous compounds, the entire Baltic Sea fails to reach good environmental status (GES), with respect to descriptor 8 and 9, as described in the Marine Framework Directive (HELCOM 2018). One significant emission source of hazardous compounds to the Baltic Sea is antifouling paints (Lagerström et al 2018; Ytreberg et al 2016). Antifouling paints are used to prevent fouling, i.e. the settlement and attachment of marine organisms such as barnacles and algae on boat hulls. The paints leach biocides into the water as a means to deter or poison fouling organisms (Almeida et al 2007). Most commonly, paints containing cuprous oxide ($Cu_2O$) are used, resulting in the emission of copper (Cu) to the marine environment (Dafforn et al. 2011). As the paints also contain zinc oxide (ZnO), added as a means to control the polishing rate of the paint, zinc (Zn) is emitted concurrently (Yebra et al 2016). Antifouling paints containing $Cu_2O$ are biocidal products and require authorization at national level to be sold within a specific country. Specific restrictions for certain regions within a country may also apply (Lagerström et al., 2018). The biocidal content of antifouling paints available on the market can therefore differ both between and within Baltic Sea States. Hence, the environmental pressure of biocides along the coastline of the Baltic Sea is a function of boat density and prevailing legislation.

General concern of air pollution is associated with human health effects, which are strongly connected to air concentration of particulate matter (PM). These small particles enter human pulmonary system and have been shown to contribute to cardiovascular diseases and childhood asthma (Lepeule, 2012; Zheng, 2015). Particulate matter is not only emitted from internal combustion engines, but it is also formed as a result of atmospheric processes. There are several other pollutants which contribute to this process, like nitrogen oxides (NOx), volatile organic compounds (VOC) and ozone. For coastal areas, waterborne traffic, and especially boats, contribute to air quality problems. However, the data and existing literature concerning the air emissions of small boats is scarce. Some studies for the spatial and temporal characteristics of recreational boating do exist (Montes et al 2018; Sidman et al 2005; Gray et al 2011), however, isolated case studies for such characteristics alone are not yet sufficient for the estimation of dynamic emission datasets on a multi-national level.

Air emission limits of leisure craft engines (EU, 2013) are significantly different from those of large marine diesel engines used in ships (IMO Marpol Annex VI, 2008). This concerns especially carbon monoxide (CO) and hydrocarbon emissions. Also, the fuel efficiency of small recreational boat engines is poor compared to large diesel engines. For example, the recommended (EEA, 2016) consumption per power unit for small boat engines can be two to five times higher than a typical marine diesel engine.

In this paper we present the first holistic approach and a model (BEAM) for the assessment of leisure boat activities and emissions for $PM_{2.5}$, $NO_x$, non-methane VOC's (NMVOC), CO and selected antifouling paint (AFP) contaminants (copper and zinc). We have used the model for leisure boats at the Baltic Sea and in our modelling approach both the temporal and spatial distribution of emissions are considered. We have utilized a wide range of information sources and data processing techniques in our modelling, including: i) AIS-data processing for non-registered marine traffic, ii) scanning of the Baltic coastline

satellite imagery iii) existing survey material for several riparian states of the Baltic Sea, iv) available information on marina
locations and sizes and v), local land use information near marinas.
Our aim in this study is to introduce the BEAM model and provide estimates for the annual leisure boat emissions for selected
pollutants, for each tiparian state and boat category separately. We also aim to address the temporal and spatial variability of
emissions and compare leisure boat emissions against the ones from the registered marine fleet. The presented modelling
approach is not exclusive to the Baltic Sea, and can be extended to other regions, given that necessary input data sets are
available.

## 2. Model formulation

In our modelling approach we assume that the whole leisure boat fleet to be modelled can be represented as a large collection
of marinas. Each of these marinas host a number of leisure boats, which are assumed to operate in the vicinity of their marinas.
Each of these marinas have a specified maximum capacity of boats they can host and the actual amount of boats at the marina
can change dynamically depending on the time of year.
To illustrate the modelling process, let us consider a selected marina with a latitude coordinate $c$ at a given hour of year $t$ with
a total amount of $N$ leisure boats at the selected marina. The amount of boats at the marina can be represented as a collection
of "bins" (a set of all possible boat types based on the generic boat class and engine setup) and the boats are distributed into
these bins according to their boat class and engine-setup. For each of these bins an "average" leisure boat can be defined to
represent all individual boats in the bin, while the nationality and location of the marina can affect the attributes of this
averaged boat. The averaged attributes include, e.g., an average travel distance per year, speed, water surface area, engine
load, installed engine power and the mix of used antifouling paint grades.
In the modelling approach all of these boat bins can be modelled independently. For simplicity let us consider a single boat
bin $i$. Let $n_i(t)$ be the amount of boats of this type that are currently situated at the marina during this hour of the year. The
amount of boats currently at the marina can be split into "active" and "inactive" boats. This split is to be done using a fraction
of activities associated to this hour $f(t,c)$, also taking into account the climatic limitations in the marina as a function latitude
($c$). The amount of active boats $A_i(t)$ and inactive boats $I_i(t)$ are given by

$$A_i(t) = \frac{N_i D_i f(t,c)}{v_i},$$
$$I_i = n_i(t) - A_i(t)$$

(1a-b)


where $N_i$ is the maximum amount of boats (at 100% capacity) at marina of type $i$, $D_i$ is the average annual travel amount for
boat type $i$, $f(t,c) \in [0,1]$ is the fraction of total activities occurring during hour $t$ and $v_i$ is the average travel distance per
hour for a boat of type $i$. In the modelling $A_i(t)$ is not restricted to be a natural integer value (e.g., values 0.1 or 1.5 can be
used) but $A_i(t)$ is required to be less or equal to $n_i(t)$, which asserts that there can be no activities in the marina in case there
are no modelled boats at the marina currently.
The assessment and geographical distribution of emissions (to air and water) that are caused by $A_i(t)$ active boats and $I_i(t)$
amount of inactive boats at the marina is modelled as follows: Inactive boats do not release exhaust emissions but contribute
to anti-fouling paint leach at a rate that is assumed to equal the rate for $A_i(t)$. The amount of fuel consumed [kg] during a
time of one hour is given by

$$FC_i(t) = A_i(t)F_{hi}$$
$$F_{hi} = SFOC_i \times P_i \times EL_i$$
(2a-b)


where $F_{hi}$ is the average unit fuel consumption, i.e., the amount of fuel a boat of type $i$ consumes during one full hour of
activity. $SFOC_i$ is the specific fuel consumption [g/kWh], $P_i$ is the average engine power rating [kW] for boat bin $i$ and $EL_i$
[0,1] is the average engine load associated with the boat class with the assumed average speed). Finally, the emission releases
for contaminant $k$ can be computed by multiplying $FC_i(t)$ with emission factors $e_{ki}$, given by

$$q_{ki}(t) = FC_i(t)e_{ki}$$
(3)


For active boats we assume that the geographic distribution of activities can be expressed with a), a finite collection of
discretely mapped locations around the marina and b), a probability distribution for these mapped locations. Then the modelled
emissions $q_{ki}(t)$ can be distributed to the mapped locations according to the distribution. The annual emission total $Q_k$ [g] is
given by

$$Q_k = \sum_{t=1}^{T} \sum_{m=1}^{M} f(t,c) \sum_{i=1}^{N} \frac{e_{ki}N_iD_iF_{hi}}{v_i}$$
(4)


where $e_{ki}$ is the emission factor for contaminant $k$ for the boat bin $i$ (of which there are $N$ in total), $T$ is the total amount of
hours per year, $m$ defines the marina of which there are $M$ in total.
The modelling of Cu and Zn released from antifouling paints differs from exhaust emission modelling. The main reason is
that both active and inactive boats act as emission sources. Secondly, the emission factors for contaminants are affected by
the geographical distribution of the marina (different paints and release rates due to salinity are applied depending on the
marina location and legislation). Finally, the emission factor, i.e. the release rates of Cu and Zn from antifouling paints, is
dependent on time – specifically on the amount of "days spent in the water" ($t_s$), which also accumulates when the boats that
are not actively being used (berthing boats at marinas). This means that the emission factor is time dependent and unique for
each boat. For a selected boat in class bin $i$ and time $t$, the hourly release rates of Cu and Zn for contaminant $k$ is given by

$$q_{ik} = a_i e_k(t_s, \mathbf{r})$$
(5)


where $a_i$ is the average water surface area for boat type $i$ and $e_k(t_s, \mathbf{r})$ is the emission factor for contaminant $k$ that depends
on the marina location ($\mathbf{r}$) as well as the amount of days spent in the water $t_s$. Given that the dynamic emission factor $e_k(t_s, \mathbf{r})$
can be pre-processed into marina and time dependent form $e_k(t, m)$ for each boat, the total annual release of contaminants is
given by

$$Q_k = \sum_{m=1}^{M} \sum_{t=1}^{T} \sum_{i=1}^{n(t)} a_i e_{ki}(t,m) \qquad (6)$$

Where the index $i$ iterates over all boats in the marina $m$ that are present during the hour $t$.

## 2.1 The BEAM model

In order to determine the leisure boat emissions based on the assumptions presented in the paper, a new simulation model for
the leisure boat activities and emissions has been developed. This model, called the "leisure boat emissions and activities
simulator" (BEAM) is illustrated in Fig. 1. In general, the model combines leisure boat characteristics, a derived temporal
profile and a geographic distribution of marinas to function. For the Baltic Sea - for which the model is being used in this
study - we utilize survey data and other available study material to characterize national leisure boat fleets and derive emission
factors for the modelled leisure boats (Sect. 2.2 and 2.3). For the assessment of general temporal profile of activities, AIS-
data for the Baltic Sea has been collected for the years of 2014-2016. Using data filters we have separated a collection of ships
from the AIS-data that exhibit the behaviour associated specifically to leisure boats; based on this filtered AIS-data we have
used the STEAM model (Jalkanen et al., 2012) to predict a temporal variation of leisure boat activities (Sect. 2.4). For the
spatial variability of activities we have compiled an extensive list of marina locations with boat count estimates. The list
includes more than 3000 marina locations at the Baltic Sea hosting approximately 250 000 leisure boats.

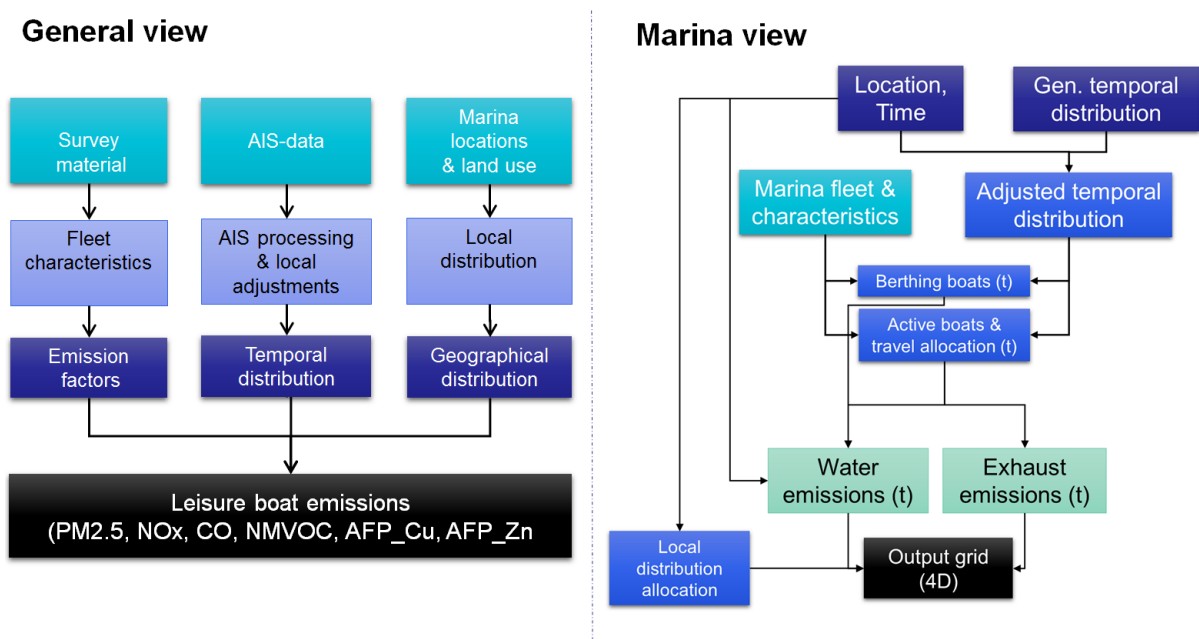

**Figure 1a-b: A diagram describing the general modelling approach for the assessment of leisure boat activities and emissions.**

The modelling approach in a more detailed overview has been illustrated in Fig. 1b. For each listed marina location the amount
of boats and their fleet characteristics are assessed. Throughout the simulation, the date of appearance and departure of each
modelled boat in marinas are being tracked, which will facilitate more realistic modelling of loads of Cu and Zn from anti-
fouling paints. The temporal profile of leisure boat activities is modelled on an hourly basis, also taking into account the
marina location and the estimated boating season length in that location. In particular, for each hour based on the activity
profile, boats are split into active and inactive (berthing) boats as in Eqs. (1a-b); the active leisure boats are simulated to
operate in the coastal area near the marina, contributing to air emissions according to their fuel consumption, engine properties
and emission factors associated to their engine setup (Sect. 2.3).

**2.2 Boat characteristics**

For the assessment of boat characteristics available information describing the leisure boat fleets for each riparian states of
the Baltic Sea were gathered. The most important source for information was survey data and existing reports based on the
surveys (Sweden (Swedish Transport Agency (2010, 2015), Germany, and Denmark), but also prior local modelling results
(Finland) and port statistics (Baltic states and Denmark) were utilized. For some riparian states (Poland, Russia) little or no
information was available to characterize the national leisure boat fleets.
The most detailed information on the characterization of national leisure boat fleet composition by far was available for the
Swedish fleet. A detailed questionnaire survey was conducted by the Swedish Transport Agency (2010, 2015). The survey
included qualitative information on the activities of 881 000 leisure boats in Sweden, including fleet characteristics, qualitative
fuel consumption and travel habits. The Swedish fleet is the largest one at the Baltic Sea and the Swedish coastline covers a
large part of the Baltic coastline, ranging from the northern parts of the sea all the way down to the southern parts of the sea.
This national study uses a 4-tier classification for leisure boats (OSB, MB, LMB and LMSB, Table 1) and for each of them
there are 5 possible engine setups, the exception being "open small boats" (OSB) which we assume are all gasoline powered.
This characterization with 18 different boat "bins" was also adopted in the BEAM model.
**Table 1: leisure boat classes and assigned attributes based on Swedish survey data (Swedish Transport Agency, 2010 and 2015).**
**For different engine setups three values have been specified in the following order: share of engine setup, the average maximum**
**engine power rating and the average engine load.**

| Description | OSB<br>Open small boat<br>(engine <7kW) | MB<br>Motorboat (engine>7kW,<br>no overnight stay) | LMB<br>Large motorboat with<br>overnight stays | LMSB<br>Large motor<br>sailing boat |
|---|---|---|---|---|
| Share of fleet | 11 % | 53 % | 22 % | 15 % |
| Water surface [m2] | 7 | 11 | 16 | 26 |
| Travel distance [km/a] | 57 | 228 | 323 | 695 |
| Average speed [km/h] | 12 | 28 | 29 | 29 |
| Older diesel engines | - | 8.7% 40kW(50%) | 21% 150kW(50%) | 40% 150kW(10%) |
| Newer diesel engines | - | 11% 40kW(50%) | 27% 150kW(50%) | 51% 150kW(10%) |
| Older Gasoline 2-stroke | 28%, 6kW(70%) | 17% 50kW(50%) | 5.4% 80kW(50%) | 1.4% 50kW(10%) |
| Newer Gasoline 2-stroke | 56% 6kW(70%) | 33% 50kW(50%) | 11% 80kW(50%) | 2.8% 50kW(10%) |
| Gasoline 4-stroke | 15% 6kW(70%) | 31% 50kW(50%) | 36% 80kW(50%) | 4.8% 50kW(10%) |

For practical modelling purposes the qualitative survey information on travelling and fuel consumption habits have been
converted into quantitative information. As an example, in the questionnaire the number of boats that report travelling 5 to 10
nautical miles per year is available, and we interpret that each of these vessels travel 7.5 nautical miles on the average (see
Appendix B for more information). To obtain average operational speed we have used the survey data that describes the
maximum operational speed multiplied with 0.7 (for LMSB this information was not available and the speed value is assumed
to equal to the one listed for LMB).
The survey also contains detailed information about engine setups (e.g., stroke type of the engine, fuel type) which was used
for all the four leisure boat classes. To calculate the fuel consumption and emissions, assumptions are needed to be made
about the effective engine load factor, i.e. the fraction of installed engine power that is used on average while the boat is
moving. There is no data available for this parameter and as a base case we have used the value in the Guidebook (EMEP/EEA
2016) of 50 %. For sailboats we have modified this value to account for that these boats do not use the engine for all travelling.
Further, for OSB the installed engine power is usually low and used with a slightly higher average engine load factor (we use
70 % instead of 50%).
The Swedish national survey data also describes some qualitative fuel consumption statistics for the boat categories but these
are not utilized directly in the modelling. Rather, we have used this information to verify that our assumptions on the key
factors that define the fuel consumption rates are in agreement with the total fuel consumption statistics derived from the
survey data (Appendix B). It should be noted that for most of the riparian states other than Sweden, such a detailed
characterization of the leisure boat fleet was not available; therefore the Swedish survey information is widely utilized in this
study also for the other riparian states, for which less information is available. The exceptions are for the Danish and German
fleet for which existing information was available for "Share of fleet" in Table 1. A description of the available information
on the fleet characteristics for other riparian states has been presented in the Appendix A.
**2.3 Emission factors and fuel consumption**
Emission factors and fuel consumption factors are given for in EMEP/EEA 2016 for different boat types, fuel type
(diesel/gasoline), and engine types (2-stroke/4-stroke engines) and emission class (divided into older conventional engines
and engines following the 2003/44 EU standard). The boat types in EMEP/EEA 2016 do not exactly overlap the boat types
used in the Swedish survey and therefore we have matched these to get usable emission factors (Table 2). It should be noted
that the emission factors of Table 2 for 2-stroke gasoline engines for CO and NMVOC are very high; for NMVOC the gasoline
engines in general have a significantly larger emission factors than the Diesel engines. Conversely, the older Diesel engines
have clearly the highest $NO_x$ emission factors.
**Table 2: Specific fuel consumption (SFOC) in g/kWh and emission factors in g / kg of fuel consumed for different boat classes and**
**engine setups. "2S" and "4S" stand for the two-stroke and four-stroke gasoline engines. "_2003" stands for the newer type of engine**
**(older type if not specified). "DSL" stands for diesel engine setups.**

| | Engine setup | SFOC (g/kWh) | PM (g/kg) | NOX (g/kg) | NMVOC (g/kg) | CO (g/kg) |
|---|---|---|---|---|---|---|
| LMSB | 2S | 791 | 12.6 | 2.5 | 322 | 539.8 |
| | 2S_2003 | 791 | 12.6 | 2.5 | 53.9 | 232.6 |
| | 4S | 426 | 0.2 | 16.4 | 50.7 | 348 |
| | DSL | 281 | 5 | 64.1 | 7.7 | 19.8 |
| | DSL_2003 | 281 | 3.6 | 34.9 | 6.7 | 18.6 |
| LMB | 2S | 791 | 12.6 | 3.8 | 215.5 | 472.8 |
| | 2S_2003 | 791 | 12.6 | 3.8 | 39.8 | 169.4 |
| | 4S | 426 | 0.2 | 28.2 | 21.1 | 293.4 |
| | DSL | 275 | 4.4 | 31.3 | 7.2 | 19.8 |
| | DSL_2003 | 275 | 3.6 | 31.3 | 6.1 | 18.6 |
| MB | 2S | 791 | 12.6 | 2.5 | 322 | 539.8 |
| | 2S_2003 | 791 | 12.6 | 2.5 | 57.5 | 232.6 |
| | 4S | 426 | 0.2 | 16.4 | 50.7 | 431.9 |
| | DSL | 281 | 5 | 64.1 | 7.7 | 19.8 |
| | DSL_2003 | 281 | 3.6 | 34.9 | 6.3 | 18.6 |
| OSB | 2S | 791 | 12.6 | 2.5 | 322 | 672.6 |
| | 2S_2003 | 791 | 12.6 | 2.5 | 57.5 | 556.3 |
| | 4S | 426 | 0.2 | 16.4 | 50.7 | 1032.9 |

For the modelling of emissions the averaged boat characteristics shown in Table 1 gives the average total annual travel
distance ($D_i$) and the average speed for Eqs. (1a-b). For each boat class and engine setup the unit fuel consumption $F_{hi}$ can be
computed based on Eq. (2b) by combining the data shown in tables 1 and 2. By combining this information with the emission
factors shown in Table 2 the emission can be computed given that the amount of active boats is known.
**2.3.1 Antifouling**
As previously mentioned, the antifouling paint market can differ between and within Baltic Sea states and Sweden has the
most restrictive antifouling legislation. In Sweden, the use of biocidal paints is completely prohibited in the Gulf of Bothnia,
and in the Baltic Proper (south of Öregrund to Trelleborg) products holding only low (5 - 8 %) $Cu_2O$ are allowed. Only on the
Swedish West Coast (North of Trelleborg), is the antifouling paint market comparable with the other Baltic Sea states, with
authorised paints holding up to 40 % $Cu_2O$. The release rates of biocides have shown to be affected by salinity, and a lower
release rate is expected in the less saline Baltic Sea as compared to fully marine waters (Ferry and Carritt, 1946, Rascio et al.
1988, Kiil et al. 2002, Adeleye et al. 2016). Recent field studies have also shown a 2-fold increase in copper release rate when
five antifouling coatings were exposed in Gothenburg (salinity, PSU, 14) as compared to when exposed in Stockholm
archipelago (salinity 5) (Lagerström et al, 2018).
**Table 3: Properties of the antifouling paints assumed to be used in the study. Data was obtained from the Swedish Chemical**
**Agency's pesticides register, from the paints' safety data sheet and technical data sheet.**

| | Antifouling paint | $Cu_2O$ (%) | ZnO (%) | Authorized usage |
|---|---|---|---|---|
| **A** | Mille Light | 6.9 | 10 – 25 | Boats> 200 kg with main mooring on the East or West coast of Sweden. |
| **B** | Biltema Baltic Sea | 7.5 | 20 – 25 | Boats> 200 kg with main mooring on the East coast of Sweden. |
| **C** | Cruiser One | 8.5 | 10 – 25 | Boats> 200 kg with main mooring on the East or West coast of Sweden. |
| **D** | Biltema West coast | 13 | 15 – 20 | Boats> 200 kg with main mooring on the West coast of Sweden. |
| **E** | Mille Xtra | 34.6 | 10 – 25 | Boats> 200 kg with main mooring on the West coast of Sweden. |


Four different geographical areas (defined in Table 4 and shown in Fig. 6) were designated here to account for the regional
differences in the antifouling paint market as well as the impact of salinity on the release rate of Cu and Zn. Release rates of
Cu and Zn from five different coatings available on the Swedish market at two salinities (5 and 14) were obtained from
Lagerström et al. (2018) as it is the only currently existing study with relevant release rates for the Baltic Sea. As it was not
possible to receive boat specific information about antifouling paint usage, it was assumed that all boats in the Baltic Sea are
painted with one of these five coatings. Hence, the BEAM model does not account for e.g. releases of restricted biocides
potentially being released from other coatings. Information about the antifouling paints used in the current study are shown
in Table 3. A salinity of either 5 or 14 was assumed for each area to determine release rates of Cu and Zn. The paint usage in
"Western Baltic", "Southern Sweden" and "Northern Sweden" were based on the Swedish regional restrictions (Table 3). For
area "Other", only paints available on the Finnish market (all but one) were considered. In Lagerström et al. 2018, the release
of Cu and Zn was studied over a time period of 84 days at various time intervals (between day 0 and day 7, 14, 28, 56 and
84). Polynomial curves were fitted to the measured cumulative release of Cu and Zn, allowing the modelling of release rates
with a daily resolution. For each geographical area, the average daily release rates of Cu and Zn from the paints was calculated
(Fig 2). In Lagerström et al. (2018), the release of Cu and Zn were only studied for up to 84 days. In addition, very thin paint
layers were used which could contribute to uncertainties in the prediction after day 56 at the higher salinity (14) as the
measured release could have been affected by the paint becoming depleted in Cu and Zn, resulting in an (erroneous) lower
release rate. After day 56, a constant release rate was therefore assumed for all geographic areas to avoid any such potential
error.
**Table 4: Geographical areas and their assumed antifouling paint use. For each area, the release rates of the used paints were**
**averaged, e.g. the release rate for Southern Sweden are based on average values from paint A, B and C. The release rate calculations**
**were based on paint-specific release rates from Lagerström et al. (2018) where these were derived from exposure at salinity 5 and**
**14. The salinity assumption for each area is also listed here.**

| Area | Geographical extent | Paints used | Salinity (PSU) |
|---|---|---|---|
| **Western Baltic** | Swedish West coast (Trelleborg to Norwegian border) and German coast (West of Stralsund) | D, E | 14 |
| **Southern Sweden** | Swedish East coast from Trelleborg to Öregrund | A, B, C | 5 |
| **Northern Sweden** | Swedish East coast from Öregrund to the Finnish border | None (prohibited) | None |
| **Other** | Coastlines of Finland, Estonia, Latvia, Lithuania, Poland and Germany (East of Stralsund) | A, B, D, E | 5 |


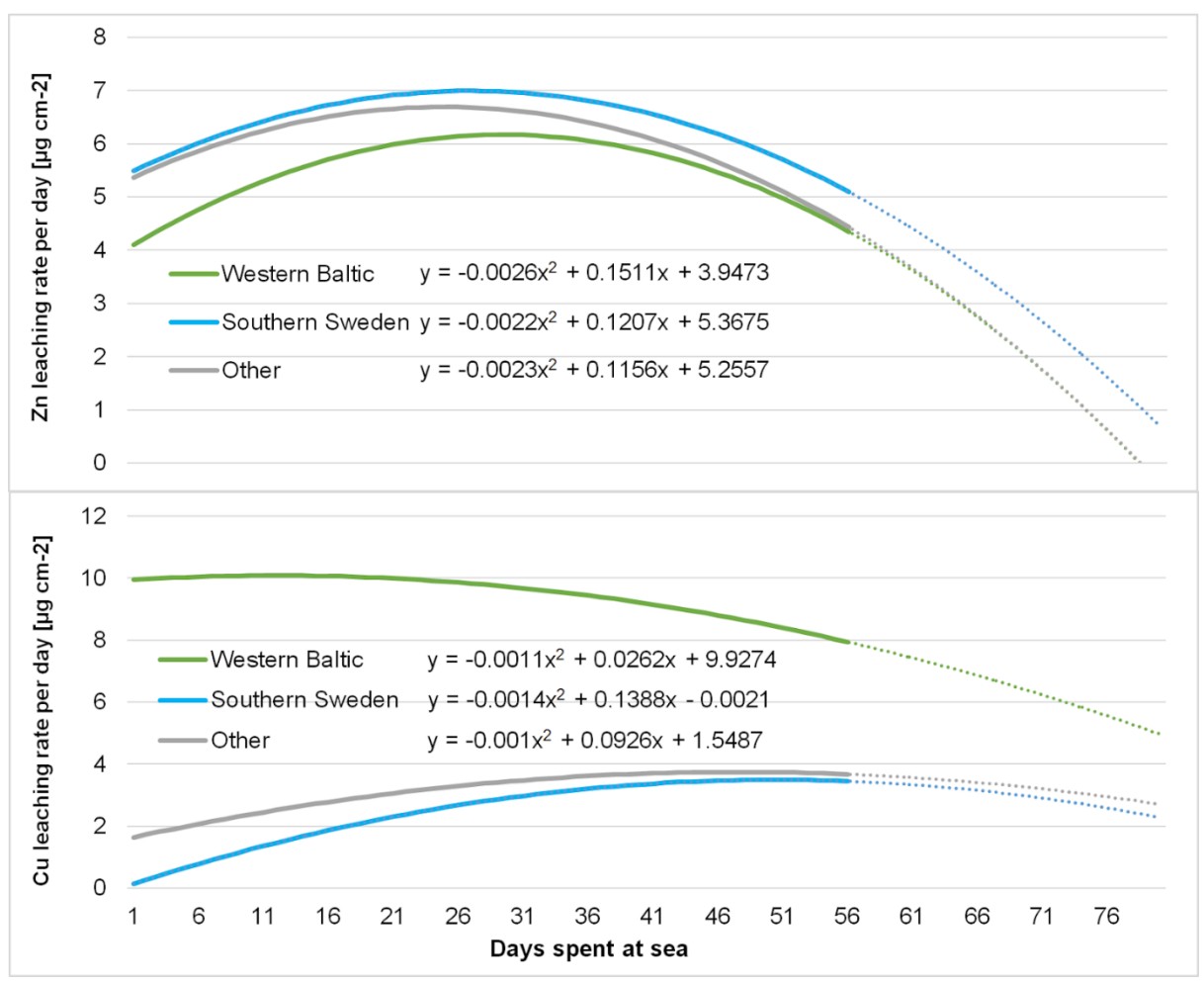

Figure 2: Cu and Zn contaminant leaching rates as a function of days spent in the water (at sea) for different areas of the Baltic Sea, as described in Lagerström et al. 2018.

**2.4 Temporal profile of activities**

AIS-transmitters are mandatory for vessels larger than 300 gross tons (GT). However while leisure boats in general are much smaller than 300gt, some boat owners (which probably represent a small subset of LMB and LMSB vessel owners) use AIS voluntarily, e.g., for safety reasons. As a consequence, while the AIS-data cannot facilitate reliable leisure boat modelling with full temporal and spatial coverage by itself, the data can still be used for the assessment of a generic temporal variation for the activities of leisure boats. In this study we have used the STEAM-model results based on AIS-data for the years 2014-2016 to identify vessels that exhibit leisure boat -like behavior. The AIS-data was given by Helcom, by the courtesy of the Baltic Sea riparian states. The identification criteria, which were designed based on the survey data for leisure boats, are as follows:

- **Seasonal activities:** The ship must be active only during the ice free season from 1st April – 30th October.
- **Low annual travel amounts:** Total travel distance for the ship must not exceed a selected threshold value of 1000 km per year.

267          -    **Small and non-commercial:** The vessel is non-IMO-registered. In case the length has been specified in static AIS-
268               data the length must be less than 15m.

269          -    **Low utilization:** The relative monthly cruising time for the ship must not exceed a selected threshold of 5 %.

Using the selection criteria described above approximately 800-1500 vessels, depending on the year, were identified. It should
be noted that the strict identification criteria may filter out some leisure boats, however, the goal is to extract a large
representative dataset and not the largest possible dataset; the risk of false-positives (non-leisure boat vessel interpreted as
one) is significantly reduced and the derived temporal profile does not require all leisure boats to be accounted for.
Using the AIS-data from these identified vessels the STEAM model was used to assess the hourly temporal profile for leisure
boat travel distances for the years of 2014, 2015 and 2016. This process has been done by modelling the travel kilometers of
the unidentified vessels with STEAM and normalizing the resulting temporal profile to sum up to 1. Also, the three annual
profiles were aligned (e.g., so that the days of week match), averaged and normalized into a single temporal profile, which
has been presented in Fig. 3.

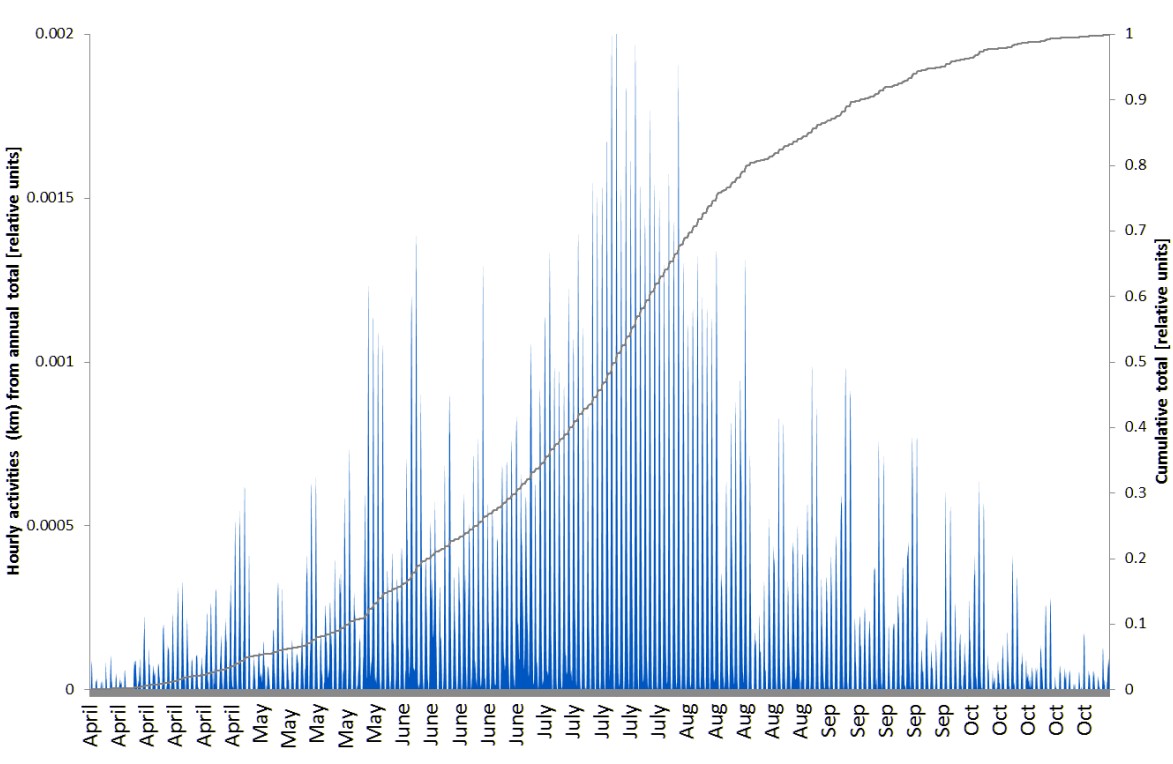

**Figure 3: Estimated hourly temporal profile (blue bars) of leisure boat activities at the Baltic Sea based on AIS-data and modelled**
**travel distances with the FMI-STEAM model. The secondary vertical axis (right) shows the cumulative temporal profile (gray line)**
**as a function of time.**
The Swedish national survey data also describes temporal patterns for leisure boat activities on a monthly basis. In Fig. 4a, a
comparison of the derived temporal profile using AIS-data is compared against the reported monthly profile given by the
survey data for all Swedish boats. The survey includes in-land use of boats, because the distinction was not made to boats
along the Baltic Sea coastline and inland water areas. Regardless, the strong correlation of these two suggests that the temporal
profile given by AIS-data can be used for the assessment of leisure boat activities. In Fig. 4b, it can be seen that the different
leisure boat categories exhibit similar temporal patterns throughout the season, based on the survey data. Thus the derived
AIS-pattern for temporal activities can be applied to each boat category without additional modifications.

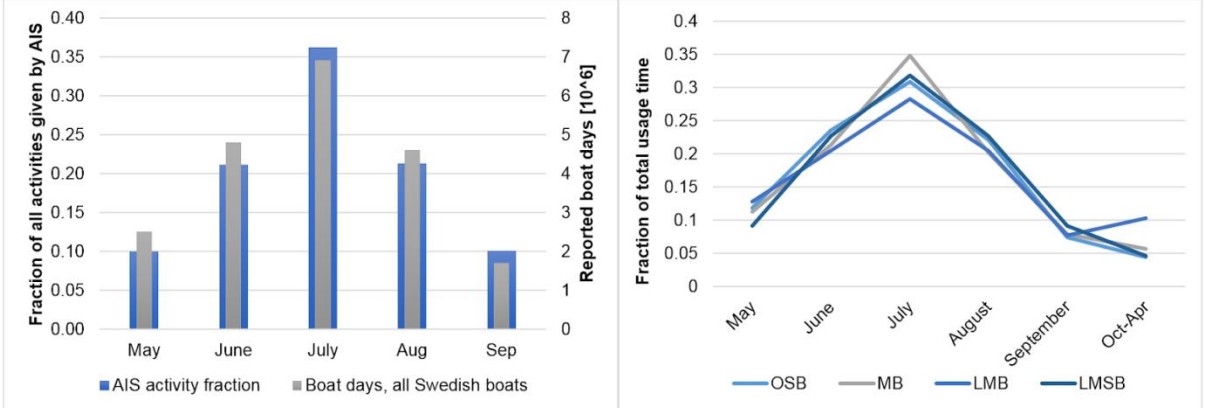

**Figure 4a-b: In a), left, comparison of the temporal profile given by AIS versus reported boat day's for all Swedish leisure boats**
**in 2010. In b), right, the reported temporal profiles for each leisure boat class separately is presented based on the reported boat**
**days for all Swedish boats.**
**2.4.1 Temporal profile adjustment**
The temporal profile based on AIS describes the leisure boat activities at the Baltic Sea in general and can be used for the
assessment of $f(t, c)$ used in Eqs. (1,4). However, this temporal profile still lacks the seasonal characteristics that occur for
different parts of the Baltic Sea. In the northern parts of the sea the boating season is shorter and starts later during the early
summer. In order to take this effect into account, a survey (interviews with marina representatives) was conducted to
investigate the temporal patterns of marinas hosting leisure boats at varying latitudes. The survey was conducted by
interviewing the marina captains of 11 marinas along the Swedish east and south coastline. The interviews were conducted
on the 14th and 16th of March 2018. The marina captains were asked to give the number of boats in their marina and assign
an approximate percentage of the marina occupancy during the boating season. This included the date when boat owners
normally start to launch their boats in spring, dates for which the marina captains could assign the marina occupancy and the
date when boat owners normally has taken up their boats from the marina. Instead of using a standardized questionnaire,
where the marina captains should assign a specific occupancy to fixed dates, the marina captains were allowed to select dates
for which they were confident to give a good estimate of the occupancy percentage. The survey results have been presented
in Fig. 5 and locations of corresponding marinas are shown in Fig. 6.

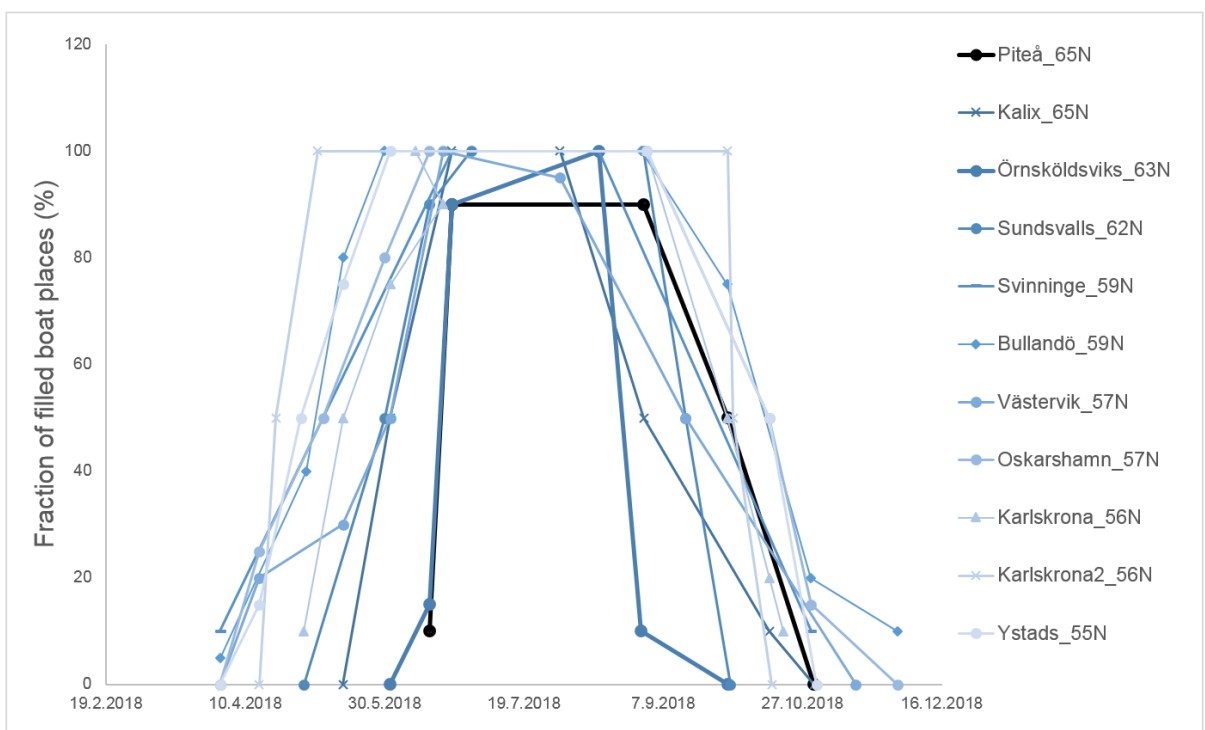

**Figure 5: Seasonal patterns based on survey data for selected marinas at the Swedish coast, indicating the utilization rate of marinas as a function of time.**

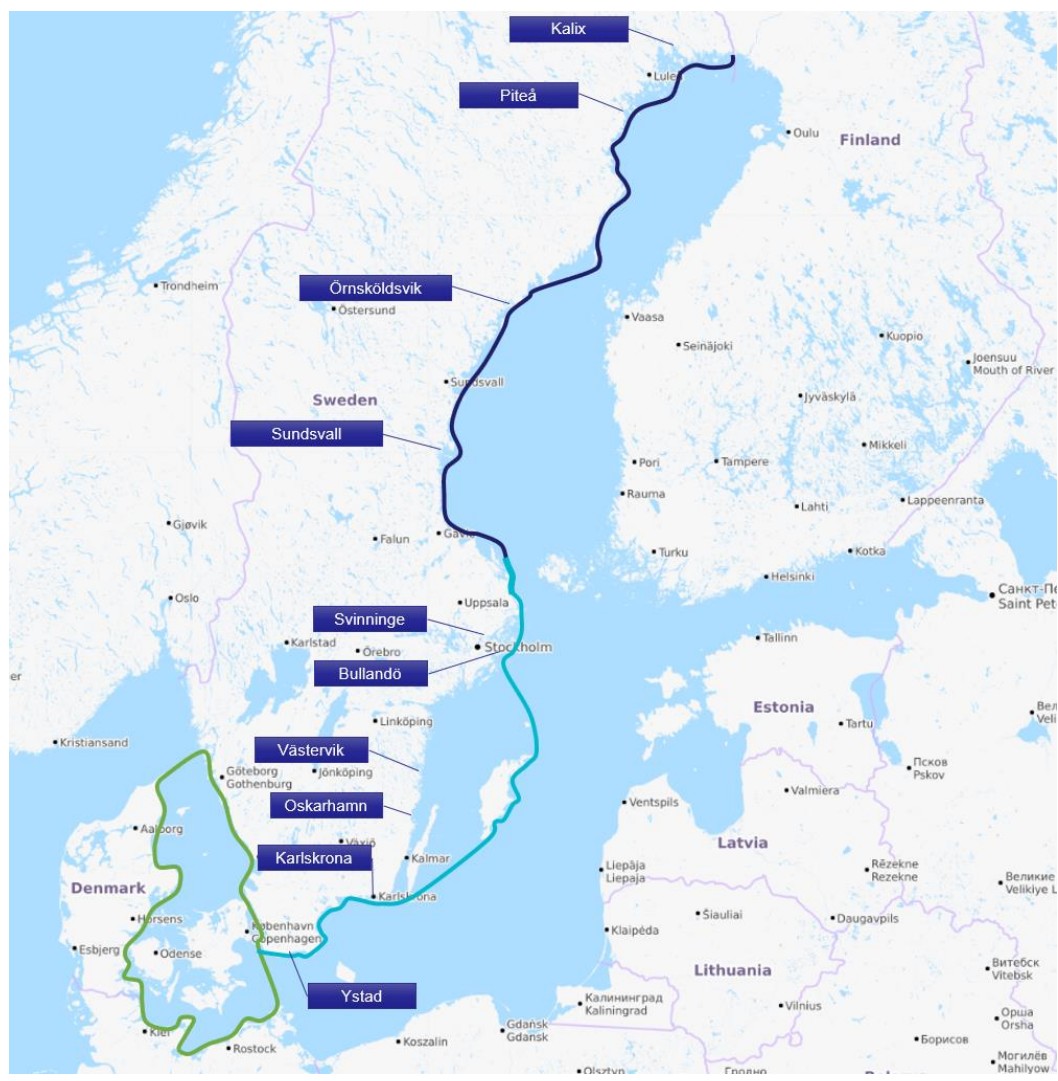

**Figure 6: Location of the Swedish boat marinas which were used to determine the boating season length in the Baltic Sea area.**
**Different antifouling paint zones have been illustrated with lines. Dark blue = "Northern Sweden". Cyan = "Southern Sweden".**
**Green: "Western Baltic". Coastal areas for which the AFP-zone has not been defined belong to the zone "Other". Map image**
**provided by © OpenStreetMap contributors 2020, distributed under a Creative Commons BY-SA License.**

Based on this survey data a simple statistical model was set up to estimate the season properties, which includes the length
($L$) and the mid-season day ($D_M$) of the season as a function of latitude coordinate in WGS84-projection ($c$). In addition, a
"ramp-up" ($L_U$) and "ramp-down" ($L_D$) length measured in days were also evaluated which describe the amount of days the
boat counts increase to 100 % and decreases down to 0 % respectively. The simple linear statistical model, which is valid at
the Baltic Sea only (53ºN < c < 66ºN), has been defined as follows:

$$D_M = 1.8c + 102,$$
$$L = 720 - 9.1c,$$
$$L_U = 0.2L, \; L_D = 0.33L$$

(7a-c)

The temporal profile adjustment has the following implications: in the northern marinas the season is shorter and starts later,
which will affect the distribution of emissions. Secondly, all activities given by the general temporal profile when no boats

are present at the marina are ignored; however we still assume the same amount of total activities regardless of latitude and therefore normalize the marina-specific profile to sum up to 1. According to the survey data and Eqs. (7), when the boat season begins ($D_M - L/2$) the marina capacity utilization reaches 100 % rapidly in 3-4 weeks; when the season ends ($D_M + L/2$) this utilization rate has decreased to 0 % in 4 -6 weeks' time. For a more concrete example, let us consider a marina near Stockholm ($c = 59.0^\circ N$). According to Eqs. (7) the length of the season is 180 days, starting around 28[th] of April and ending around the 27th of October. The "ramp-up" amount of days is estimated to be 36 days and therefore by 3[rd] of July the marina is expected to have reached 100 % capacity. After the beginning of September the capacity gradually starts to decrease and after approx. 60 days the marina is expected to be empty until the season starts again next year.

**2.5 Geographical distribution of boats and activities**

The modelled geographical distribution of leisure boat activities is a product of two separate processes: first, the list of marina locations with boat count estimates will outline the general geographical distribution at the Baltic Sea. Secondly, at the vicinity of each marina location the boat activities are allocated on a higher resolution, taking into account land cover information. A list of leisure boat harbours (boat place counts, location) for each riparian state was collected based on survey data, existing national studies and satellite image analysis. The satellite image analysis for marina locations and sizes was performed manually (Fig. 7a). However, for the Swedish coastline a digital mapping of the marinas provided by the Swedish EPA was used that described the geographic areas of the marinas; these areas were converted into boat count estimates, which were manually verified with satellite image analysis.

The full list of marinas includes more than 3000 locations for leisure boats at the Baltic and accounts for more than 250 000 boats in total. Based on the Swedish survey data 37 % of boat owners report using offshore facilities and trailers to harbor their boats. Additionally, a significant fraction of the boats are located on private shores outside of marinas. To take this into account in the modelling we assume that the listed marina locations are expected to harbour only 50 % of the total fleet and we therefore multiply each marina boat count with a factor of 2; in other words, we assume that for each boat in a marina there is another boat not accounted for and its activities can be associated to the area near the marina location. This assumption is consistent, for example with the estimates of Daehne et al, (2017), who report 43 000 German boats for the Baltic Sea coastline. Our boat count based on satellite images yields 19 900 boats for the German Baltic Sea area, but become consistent with Daehne et al (2017) estimate if offshore locations are considered.

In Table 5 the amount of boats in marinas, private shores and offshore facilities for all riparian states are shown. Also the fleet composition has been shown in the table, which has been assessed based on survey data. For the Finnish fleet it should be noted that total surveyed boat count (195000) is for fuel consuming boats without distinction between the Baltic Sea and inland waters. In another study by the Finnish authorities[1] it has been estimated that there are over 90000 boats at the Finnish coast, which is consistent with our estimates based on satellite analysis – once the multiplication with a factor of 2 is done.

---

[1] A report "Antifouling valmisteiden ympäristöriskinhallinta ja kestävä käyttö" by TUKES written in Finnish is available at: https://tukes.fi/tietoa-tukesista/materiaalit/biosidit/

Table 5: National leisure boat counts based on survey data (Appendix A) and coastal satellite image analysis. Total modelled amount of boats equals the preliminary boat count (marina) added with the estimates for boats at private shores and in trailers. The described fleet composition corresponds to the percentages used for SB, MB, LMB and LMSB types, in the given order summing to 100%.

| Riparian state | Boats, marina | Boats, private shore | Boats, offshore/trailer | Total Modelled | Total survey | Fleet type composition [%] |
|---|---|---|---|---|---|---|
| Sweden | 113900 | 84286 | 29614 | 227800 | 231900 | 11, 53, 22, 15 |
| Finland | 50600 | 37444 | 13156 | 101200 | 195000,90000 | 11, 53, 22, 15 |
| Denmark | 59600 | 44104 | 15496 | 119200 | - | 10, 34, 22, 34 |
| Germany | 20000 | 14800 | 5200 | 40000 | 42700 | 10, 15, 20, 55 |
| Russia | 2450 | 1813 | 637 | 4900 | - | 11, 53, 22, 15 |
| Estonia | 2330 | 1724 | 606 | 4660 | - | 11, 53, 22, 15 |
| Poland | 1720 | 1273 | 447 | 3440 | - | 11, 53, 22, 15 |
| Latvia | 1080 | 799 | 281 | 2160 | - | 11, 53, 22, 15 |
| Lithuania | 685 | 507 | 178 | 1370 | - | 11, 53, 22, 15 |

### 2.5.1 Local geographical distribution

The Swedish surveys (Båtlivsundersökningen, 2010 and 2015) indicate that the clear majority of leisure boats regardless of type operate very locally near their marinas. Based on this information it is sufficient for the scope of this study to allocate all leisure boat activities in the vicinity of marinas, although some marina-to-marina activities for the larger boat classes will be misallocated in this estimation. The overall process of analyzing the coastline for marina locations with boat counts and deriving local distributions of activities is described in Fig. 7.

For each marina we form a list of local discrete locations for possible boating activities defined with a selected resolution of 0.2 km x 0.2 km. The maximum range for this mapping has been set to (50 km) and land-use data has been used to omit all discrete locations that are not located at the Baltic Sea. For each of these locations the distances to the marina ($r_m$) and to the nearest coastline ($r_c$) is evaluated. For $r_c$ in particular, land-use information (OpenStreetMap) has been used in the assessment also taking into account islands.

For each listed discrete location for possible boating activities, we compute an activity probability $p(r_m, r_c)$. In this study we have opted for a simple exponential function to express $p(r_m, r_c)$, given by

$$p(r_m, r_c) = e^{-a(r_m + br_c)} \tag{8}$$

where the term $r_m + br_c$ can be regarded as the "effective" distance from the marina that also considers the distance to the nearest coastline, and the factor a defines how strongly the probability decreases as a function of this distance. Presumably the factors $a$ and $b$ depend on the leisure boat class, for example, the larger boats are used for longer travels and can safely be operated farther away from the coastline. However, due to lack of usable data we have settled for generalized empirical values for $a$ and $b$ that are the same for each boat class. Finally we normalize the probabilities so that they sum up to 1.

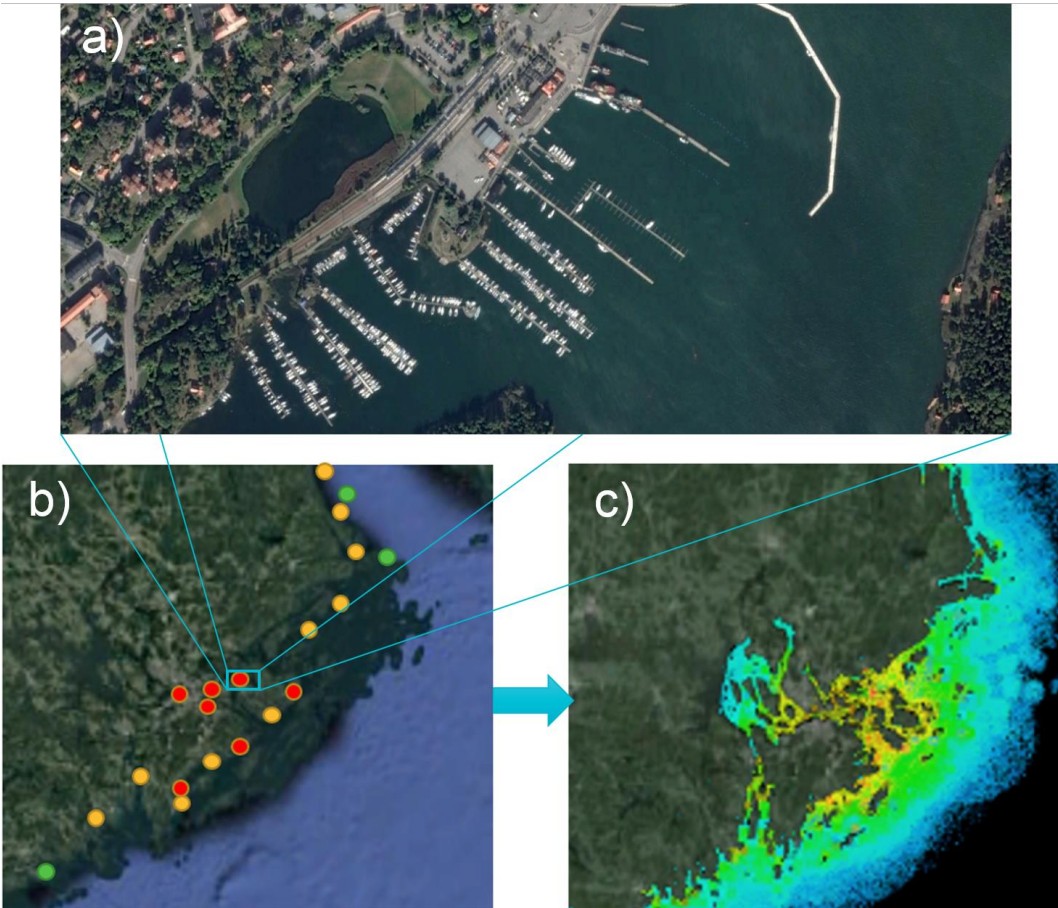

**Figure 7a-c: The overall process for geographical distribution of activities. In a), an example of satellite imagery used to calculate the number of boat places in a small boat marina (58.9 N,17.95 E, Sweden). Iterating over the analysis of a) a complete mapping of marinas and boat counts is formed (b). In c) an example is given when emissions of selected marinas have been allocated according to Eq. 8. Satellite image background provided by © Google Earth.**

In the study by (Montes at al, 2018) the distribution and intensity of recreational boating in the South-East US has been presented. The findings of the study performed in Florida are not fully applicable to leisure boats at the Baltic, however, the observed boating patterns do exhibit clear dependencies to both the coastline distance and the marina distance. Due to the lack of data we assume the effects of $r_m$ and $r_c$ to be equal and set value for $b$ to 1 and use a value of 0.2 for $a$, which we estimate to lead to similar distributions than was obtained with the generalized additive model (GAM) in (Montes at al, 2018).

### 3. Results

The BEAM model was used to estimate the hourly emissions of leisure boats at the Baltic Sea starting from 1st of March until the end of November. It should be noted that in the production of input datasets a heterogeneous collection of survey material, AIS-data, and satellite imagery was utilized dating between 2010 and 2017. Therefore the presented results do not represent any specific year in particular. The modelled annual total emissions, fuel consumption and travel amounts have been presented in Table 6.

**Table 6: Modelled leisure boat fuel consumption, emissions and travel distances at the Baltic Sea, for different flag states and boat types. Flag state category "Other" includes Russia, Estonia, Latvia, Lithuania and Poland.**


| | Gasoline [10³kg] | Diesel [10³kg] | CO [10³kg] | NMVOC [10³kg] | NOx 10³kg] | PM$_{2.5}$ [10³kg] | AFP$_{Cu}$ [10³kg] | AFP$_{Zn}$ [10³kg] | Travel [10⁶ km] |
|---|---|---|---|---|---|---|---|---|---|
| **All boats** | 37800 | 21800 | 13200 | 3930 | 1220 | 400 | 57 | 49 | 162 |
| **Sweden** | 19300 | 8920 | 6750 | 2030 | 520 | 198 | 15.9 | 18.6 | 67 |
| **Finland** | 8620 | 3990 | 3010 | 904 | 232 | 88 | 5.5 | 8.4 | 30 |
| **Denmark** | 6890 | 5600 | 2410 | 694 | 298 | 78 | 23.8 | 13.7 | 41 |
| **Germany** | 1610 | 2620 | 577 | 159 | 128 | 23 | 10.3 | 6.3 | 19 |
| **Other** | 1390 | 634 | 487 | 146 | 37 | 14 | 1.1 | 1.6 | 4.8 |
| **OSB** | 739 | 0 | 468 | 102 | 3 | 8 | 2.6 | 2.3 | 2.9 |
| **MB** | 23700 | 2040 | 8560 | 2910 | 239 | 234 | 15.7 | 15.7 | 51 |
| **LMB** | 12900 | 11600 | 3860 | 801 | 582 | 121 | 12.2 | 10.9 | 34 |
| **LMSB** | 569 | 8150 | 349 | 120 | 392 | 39 | 26.3 | 19.6 | 74 |


According to the results, almost half of the gasoline fuel consumption, CO-, NMVOC- and PM$_{2.5}$ emissions comes from the
Swedish leisure boat fleet. For exhaust emissions and fuel consumptions Denmark and Finland have the second and third
largest contribution, in changing order depending on the pollutant type; Germany has the 4th largest contribution for these
estimates. Together these 4 flag states contribute 96 % - 99 % of exhaust emissions from all leisure boats at the Baltic Sea.
The combined fuel consumption is estimated to be approximately 60 million kg of which the clear majority of this is gasoline
fuel.
Quantitative estimates for Swedish leisure boat fuel consumption for all boat classes can be derived from the Swedish leisure
boat survey. Based on these survey material estimates, the modelled fuel consumption for both gasoline and diesel are in fair
agreement with modelled values (Appendix B). This agreement gives an indication that the used average speeds, engine loads
and engine power ratings can be considered realistic at least for the Swedish fleet. Emissions of NO$_X$, PM$_{2.5}$ and CO for the
whole Swedish leisure boat fleet (also comprising boats in inland waters) have been determined by the Swedish EPA for the
year 2018 and was 1273 t (NO$_x$), 148 t (PM$_{2.5}$ ), 2744 t (NMVOC) and 18854 t (CO) (Swedish EPA, 2018). Based on the
Swedish survey data, clear majority of MB and LMSB boats and half of the LMB boats operate at the Baltic Sea for the
Swedish fleet; based on this, the emission totals given by Swedish EPA seem higher for NO$_x$ and CO while PM$_{2.5}$ is lower
than the presented BEAM predictions would suggest.
The boat class –specific emission totals shown in Table 6 show that the motorboats (MB) are responsible for 74 % of released
NMVOC emissions, mainly due to high amount of gasoline used with old 2-stroke gasoline engines. The motorboats are also
modelled to be responsible for 65 % of CO emissions and 58 % of PM$_{2.5}$ emissions. Together, LMB and LMSB release 80 %
of NO$_x$ emissions due to the higher use of diesel fuel. For all modelled emission types the smallest boat category OSB has
very low shares in general.
The loads of Cu and Zn from antifouling paint are affected by sea salinity as well as the types of paint allowed and used in
the different parts of the Baltic Sea. As a consequence, 58 % of the Cu emissions and 42 % of Zn originate from the relatively
short combined coastline of Denmark and Germany due to the combination of the higher ambient salinity (resulting in higher
leaching rates) and types of paints allowed. In contrast, the much longer combined coastline of Finland, Russia, the Baltic
States and Poland produces only 11 % of Cu and 20 % of Zn emissions. For antifouling paint contaminants the contribution
from LMSB boats is the largest (26.3 t), due to the large water surface area (26 m$^2$) for this boat type.

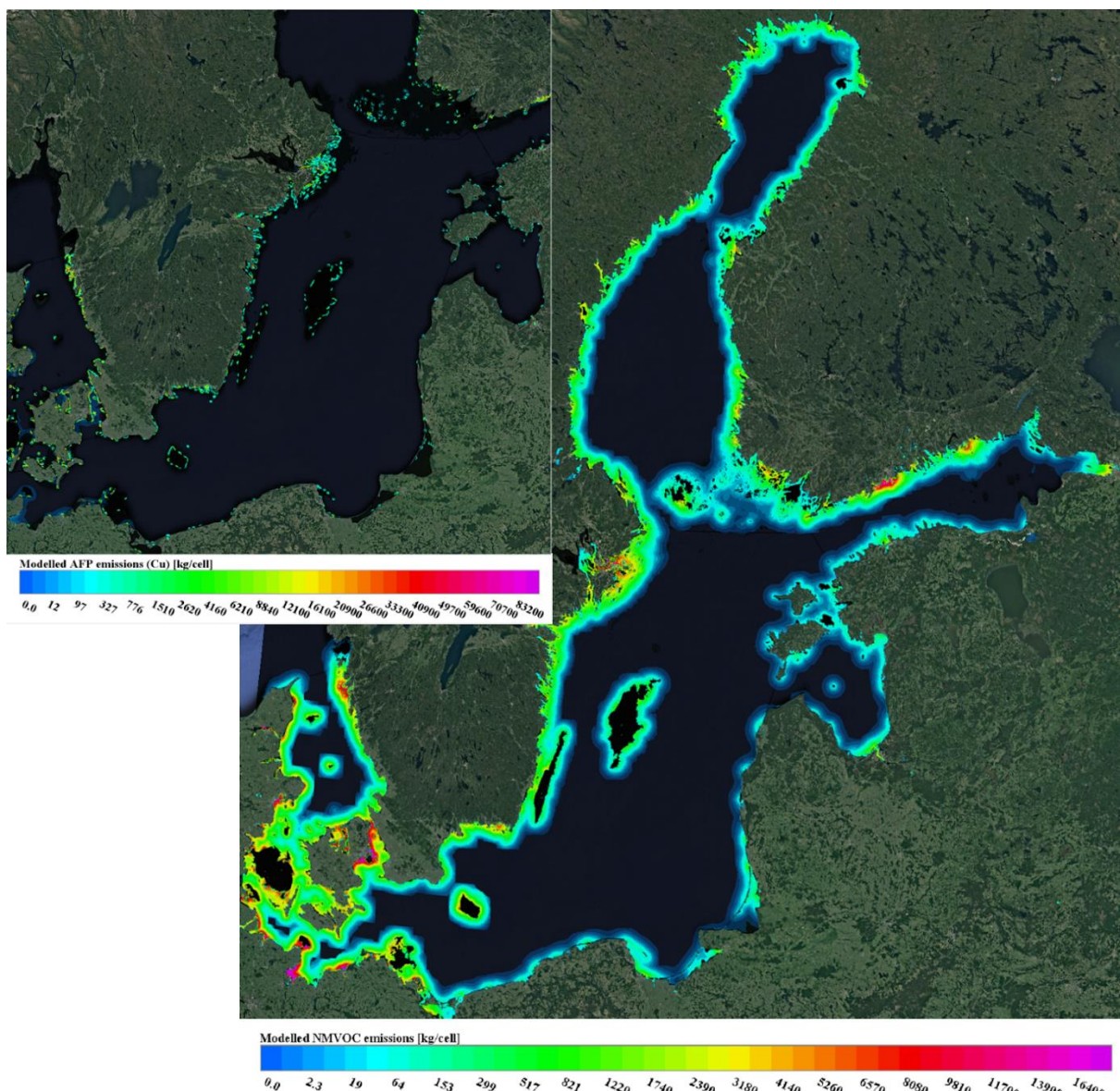

**Figure 8: Estimated geographical distribution of NMVOC exhaust emissions and the copper emissions from antifouling paints for**
**a selected area. Satellite image background provided by © Google Earth.**
In Fig. 8 the estimated NMVOC emissions have been presented. It can be seen from the figure that there are several hotspots,
including the archipelago near Stockholm, Helsinki area, Copenhagen, Gothenburg and Lubeck area. It should be noted that
the modelled geographic distribution of emissions on a local level is only indicative due to the lack of usable data to
parametrize Eq. 8. The copper emission from antifouling paints have also been presented in the figure (upper left) for the
South-Western part of the Baltic Sea and in contrast to NMVOC, the copper emissions are heavily concentrated on marina
locations. The reason for this is that all boat classes are expected to have very low amount of active hours per year, which in
turn causes the main source of releases to be stationary boats at the marina locations. As an example, consider OSB's that
have an average annual travel amount of only 57 km which is reached with less than 5 hours of activity during the year.
The geographical distribution of exhaust emissions is difficult to predict due to the lack of activity data available for leisure
boats. As was discussed in Sect. 2.4 we used AIS-data for 2014-2016 and the STEAM model to isolate a small subset of
leisure boats, which were used to assess the temporal distribution of activities. Presumably, this set of boats is a subset of the
larger boat classes (LMB, LMSB) and the geographical distribution of modelled fuel consumption for these boats should be
comparable to BEAM predictions for the largest boat classes. We used this AIS-data to model the fuel consumption of these
small number of boats for 2014-2016 and the averaged results of this modelling have been presented in Fig. 9. For comparison,
the BEAM modelled fuel consumption for LMSB-boats have been presented in the figure. To be able to compare these
modelled distributions, the modelling resolution has been set identical and the grid cell values have been scaled to be in
proportion to the average grid cell content. It can be seen from the figure that the AIS-driven approach show marina-to-marina
boat activities which have not been considered in BEAM. In both approaches the clear majority of activities (travelling amount
and thereby fuel consumption) coincide near coastal areas and are heavily concentrated on the same hot-spots - the area near
Lubeck being an exception.

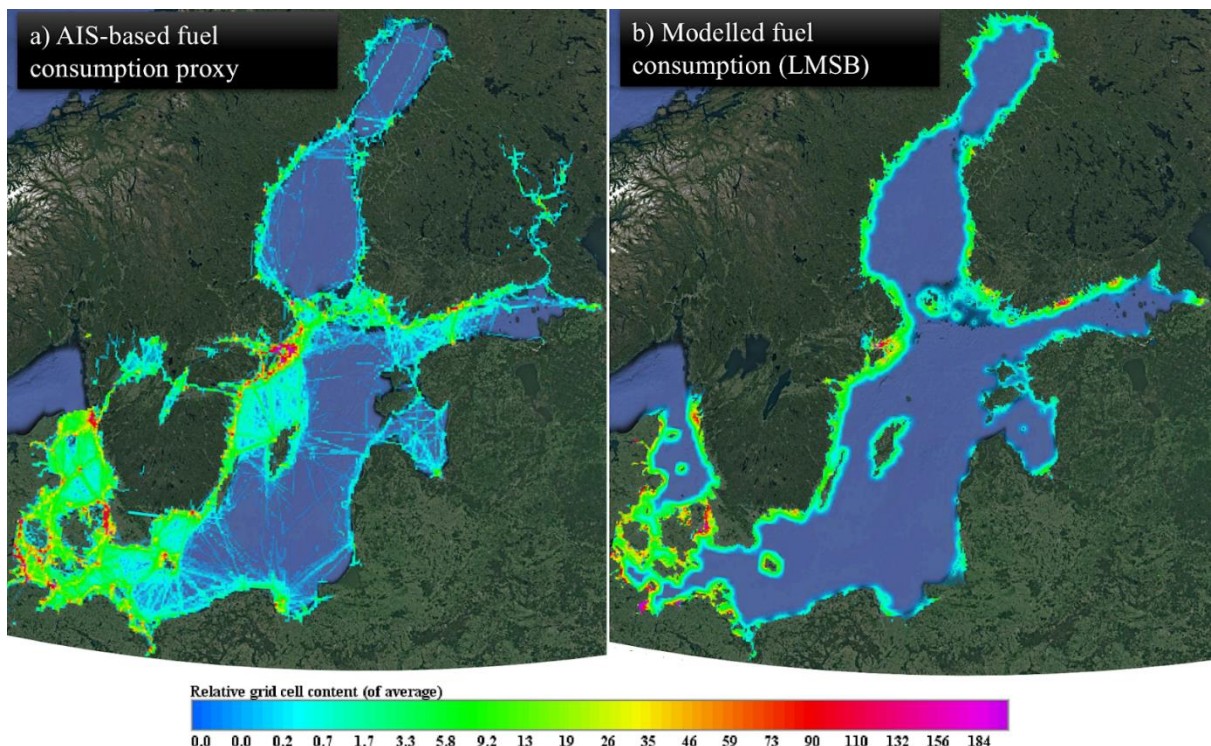

**Figure 9: Estimated distribution of leisure boat fuel consumption in terms of grid cell average. In a) the predictions based on AIS**
**(STEAM) are shown. In b) BEAM predictions for LMSB fuel consumption has been shown. Satellite image background provided**
**by © Google Earth.**

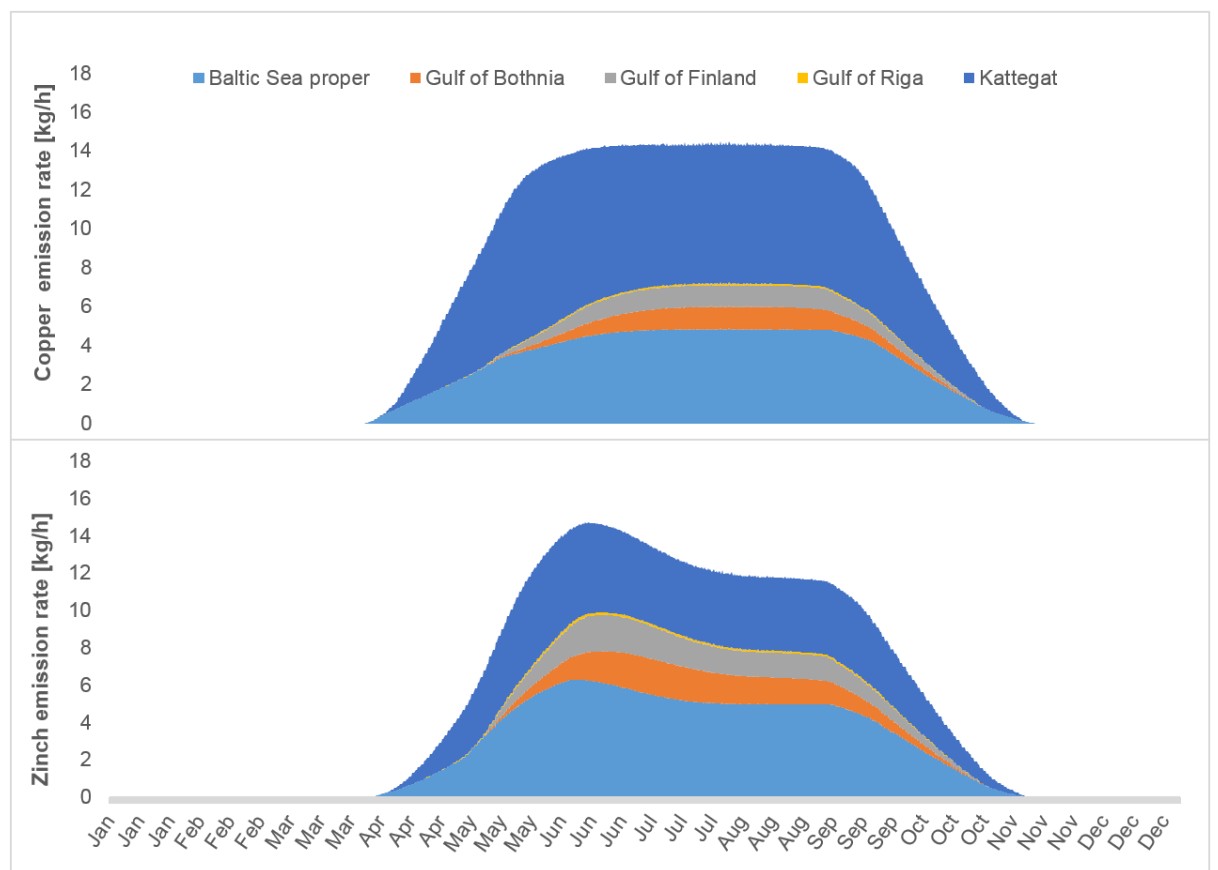

**Figure 10: Modelled antifouling paint contaminant leaching rates for different parts of the Baltic Sea.**
The modelled emissions have a clear seasonal patterns and this is especially evident for the modelled antifouling paints. The
hourly emission rates for copper and zinc contaminant is presented in Fig. 10 for different parts of the Baltic Sea. As it can
be seen from the figure, copper and zinc emissions have different temporal patterns which are caused by the dynamic emission
factors shown in Fig. 2 (Lagerström et al., 2018). According to the model, the total zinc emission release rate peaks at the
beginning of June whereas the releases of copper are more evenly distributed during the season. The two dominant regions
for copper and zinc emissions are clearly Kattegat and the Baltic Sea proper.
**3.1 Leisure boat emissions versus commercial shipping**
The emissions and impacts of registered shipping at the Baltic have been studied thoroughly, and these known emissions have
been compared against the leisure boat emissions to gain a better perspective on the presented total emissions. In Fig. 11, the
total emissions of registered shipping in 2014 using the STEAM-model have been presented. For commercial shipping, the
activities and emissions are somewhat evenly distributed during the year whereas leisure boat emissions are heavily
concentrated on the summer months. To highlight this contrast, the leisure boat emissions have also been compared against
the commercial emissions in July.
From this annual comparison it can be seen that while the total travel kilometers of leisure boats are comparable to the total
of the registered fleet, the fuel consumption, $NO_x$ and $PM_{2.5}$ is significantly lower (1.2 %, 0.4 % and 2.7 % respectively) for
leisure boats. However, the zinc contaminants from the antifouling paints and CO are lower but comparable to those of the

registered fleet and copper contaminants are 19 % from the respective total. The higher loads of copper and zinc from the commercial fleet can primarily be explained by legislation and use pattern. Paints for commercial ships are allowed to have a higher release rate of copper and zinc as compared to paints for the leisure boat market and the leisure boats are assumed to be used during April to October only. Since NMVOC emission factors for leisure boat engines are 1-2 orders of magnitude larger than for the large well-optimized marine diesel engines, the NMVOC-emissions from the leisure boats is estimated to be significantly larger than the emissions from registered vessels. In July the relative importance of leisure boat emissions with respect to the commercial fleet is greatly emphasized in this comparison. In particular NMVOC's are 500 % larger, CO emissions are 140 % larger and zinc emissions are 80 % larger for leisure boats than the emissions from registered traffic during July. It should also be noted that the emissions released by leisure boats are heavily concentrated near populated coastal areas, for which Fig. 8 gives an indication.

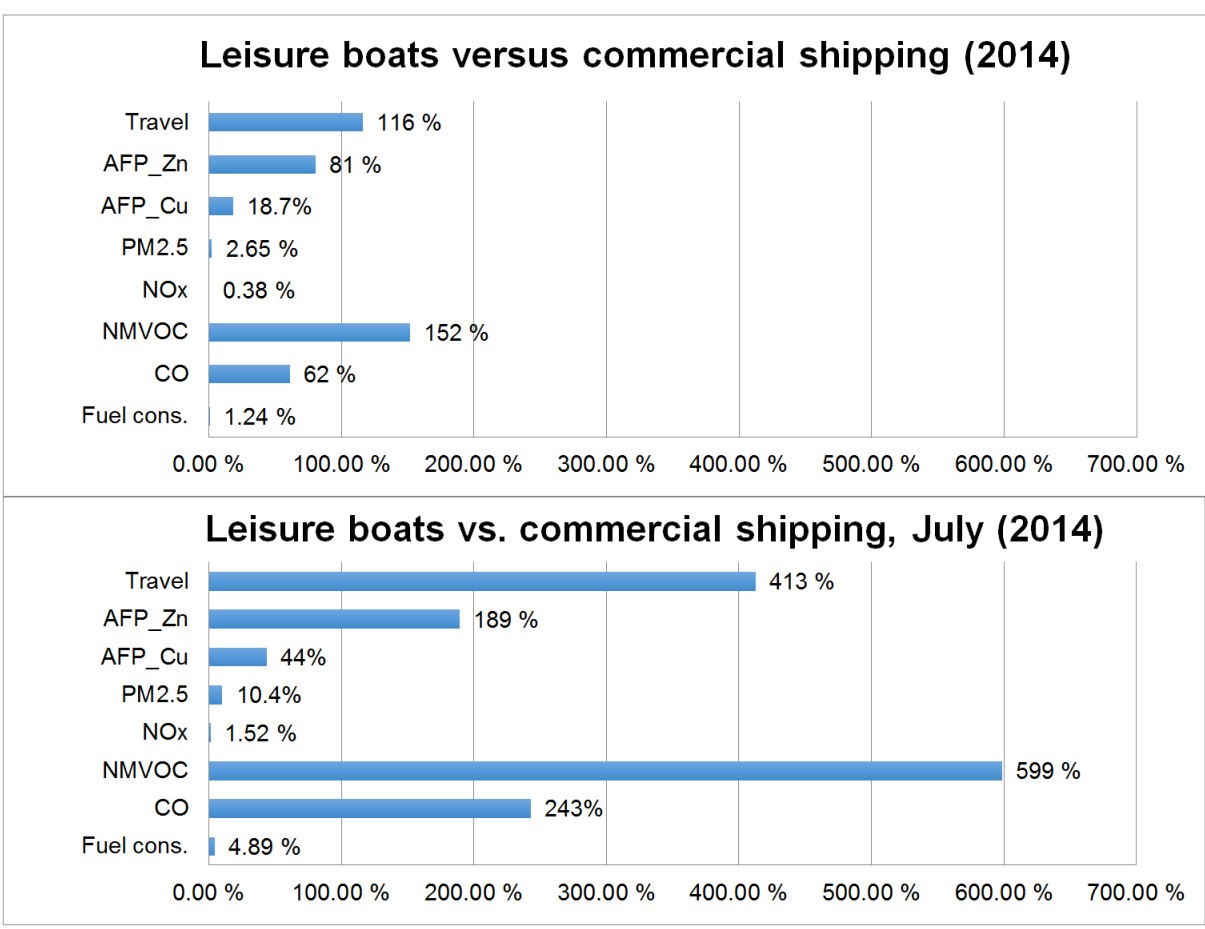

**Figure 11: Modelled leisure boat emissions, fuel consumption [kg] and travel kilometers with respect to those of the commercial fleet in 2014, and separately for July 2014. Value of 100 % indicates equal contribution from small boats and commercial shipping.**

## 4. Conclusions

A new simulation model for the assessment of leisure boat activities and emissions at the Baltic Sea (BEAM) has been presented. In the model both the temporal and spatial distribution of emissions is considered and leisure boat fleet characteristics can be customized, e.g., according to available survey material. For this study at the Baltic Sea we have utilized

a wide range of information sources and data processing techniques in our modelling, including AIS-data, coastline satellite
imagery, survey material, data on marina locations with boat counts as well as land-use information.
The leisure boat emissions have previously been largely unknown at the Baltic Sea and the results given by the presented
model improves this situation. Leisure boat emissions, being heavily concentrated on the populated urban areas during the
summer months, are rarely used – and most often neglected - in dispersion modelling or other impact assessment modelling
work such as marine ecosystem modelling (e.g. Raudsepp et al., 2019). The presented model can be used to produce dynamic
emission datasets for selected exhaust pollutants and water contaminants, which could be utilized in the above-mentioned
studies in the future.
According to our results some of the pollutants emitted by leisure boats are very substantial when compared against the
emissions originating from registered, commercial shipping activities at the Baltic Sea. This comparison has been made based
on modelled shipping emissions for 2014, however, it should be noted that the modelled emissions are fairly similar for other
years during 2012 – 2018 given by the STEAM model. CO emissions equal 70 % of the registered shipping emissions and
NMVOC emissions equal 160 % with respect to commercial shipping. However, modelled $NO_x$ and $PM_{2.5}$ from leisure boats
are clearly less significant with respect to the registered shipping emissions. In absolute terms the modelled emissions are
13000 t for CO, 3900 t for NMVOC, 400 t for $PM_{2.5}$, 1200 t for $NO_x$, 57 t for copper- and 49 t for zinc water contaminants. It
should be noted that most of the modelled emissions occur during the summer months, during which their relevance nearby
marina areas are further increased. Given the relatively large emission estimates for leisure boats, especially for NMVOC,
this commonly overlooked source of emissions deserves to be further investigated in greater detail. Also the impact on air
quality should be studied further with measurements and dispersion modeling. It should be noted that while the leisure boat
emissions are significant with respect the commercial fleet, the modelled exhaust emissions are still fairly small when
compared against, e.g., national total anthropogenic emissions. For example, the reported NMVOC emissions for Finland are
reported[2] to be 88 t in 2016.
Clear majority of the emissions can be attributed to Swedish, Finnish and Danish boats, of which the main contribution
originates from motor boats (MB, LMB), but the leisure boat fleets have the potential to become larger in Russia, Estonia,
Latvia, Lithuania and Poland in the future. The motorboats are especially dominant in the emissions of NMVOC, CO
emissions, largely due to the large amount of motorboats with old 2-stroke gasoline engines. Therefore, an effective approach
to reduce leisure boat impacts on marina areas could be to reduce the amount of these engine setups, however, a more thorough
impact analysis should be conducted first. As older engines are replaced by newer combustion engines or electrical engines
the situation will naturally improve. For anti-fouling paint leach the largest boat category has the highest impact. The smallest
contribution in terms of exhaust and water emissions comes from the smallest boat type, OSB.
The uncertainty margin for the presented results is fairly high, which should be narrowed down with further development and
research. The most notable sources for error in this study are arguably the national total boat counts. The used satellite analysis
is difficult to conduct and is subject to interpretation. Also, the boats on private shores and the use of offshore trailers is a
matter of concern for the modelling and the survey material gives only an indication on the amount of boats that are outside

---

[2] Reported Finnish emissions can be seen from: https://www.ymparisto.fi/fi-FI/Kartat_ja_tilastot/Ilman_epapuhtauksien_paastot, visited 18.12.2019.

marinas. As a second source of error, the fleet composition and the split of engine setups are difficult to customize for other riparian states of the Baltic Sea besides Sweden, which has by far the highest quality survey material available. Third, the used emission factors for different engines are based on averaged Swedish data compiled several years, which introduces uncertainties. Fourth source of error is that our treatment of temporal and spatial patterns for boats does not consider the different boat classes independently, but a generalized profile is used. Finally, the geographical distribution of activities on a local scale is based on model that was set up with very low amount of evaluation data and marina-to-marina activities could not considered. Arguably the highest confidence can be given to the temporal profile of activities based on AIS-data analysis. Even for the temporal profile further model development is required so that a specific year can targeted and, e.g., take into account the impact of weather.

**5. Appendices**

**Appendix A: Fleet description for riparian states other than Sweden**

**Fleet characteristics for Finland**

Study conducted by VTT (Finnish Maritime Administration, 2005) concluded that in Finland, there was about 390 000 small boats with a motor. The national small boat registry (Finnish Transport Safety Agency, 2015) lists over 195 000 small boats powered by an engine, which is about half of the previous assessment. The discrepancy of boat numbers may be partly because new boat registry requires an active registration of all boats with an engine. If a boat is not actively used, it may not be included in the small boat registry. The information contained in the small boat registry is only an indication of the total fleet of boats, because it does not distinguish between boats used in Baltic Sea coastline and those used in inland waters. For this reason, the satellite imagery from the Finnish coastline was searched for small boat marina locations. Vessel counting was done based on available places for boats, not the actual boats themselves, because it was likely that some of the boats were in use during the time satellite image was taken. On the other hand, counting the boat places automatically assumes 100 % usage of available capacity. Regardless, 475 boat marinas were found in the Finnish coastline, Turku archipelago area and Ahvenanmaa islands. These marinas had space for over 50 600 vessels. The vessel characteristics for the Finnish fleet are based on the Swedish survey. No official records exist for fuel sold to small boats in Finland.

**Fleet characteristics for Denmark**

The Danish Maritime Authority registers leisure boats over 20GT (type of boat, type of propellant or homeport is not available), but no register of all Danish leisure boats was seen to be available. Based on the information found from http://www.sejlnet.dk/havneguide it was possible to locate 338 marinas in the Danish part of the Baltic Sea. This information included the geo-reference of the marina, the number of mooring places as well as contact details for further contact to individual harbours. The largest of the harbours were contacted by telephone and email to inquire whether the marina had a separate fuel station, so that an estimation of fuel consumption could be made. 15 of the biggest marinas fit the conditions and were able to inform the fuel consumption in their marina, although the fuel consumption statistics were not ultimately utilized in this study. It was estimated that the listed marinas had space for over 59 600 vessels according to satellite image analysis. The fleet composition was observed to be different than the one reported for the Swedish fleet.

**Fleet characteristics for Germany**

A telephone survey formed the basis of the bottom-up statistic, which was used to obtain information on the annual fuel sales in liters (diesel and petrol) from the German water petrol stations in the Baltic Sea. All 39 petrol stations on the German Baltic Sea coast were contacted and information from 35 stations was received and recorded. Research was conducted prior to the survey on the number of water petrol stations and their connected berth.

The number of berths is relevant to the research, since the study examined whether the revenue per berth is similar in different regions on the German Baltic Sea coast and can therefore be transferred to other regions. Since there are no exact statistics on how many German leisure boats exist in the Baltic Sea, this study relates to a previous study which equates the number of berths to the number of existing vessels.

An online questionnaire formed the basis of the top-down statistics. The survey was conducted by 265 German leisure boat owners who sail the Baltic Sea. The survey asked technical questions regarding the characteristics of their vessel, such as motor and fuel consumption, as well as information on their activities. Activities were divided into two categories: popular short trips and popular long trips. The boats in the survey were classified into three sub-types: sailing boats with engines, sailing boats without engines, and motorboats.

It was estimated that the listed marinas had space for approx. 20 000 vessels according to satellite image analysis, which was approximately half of the amount of vessels according to survey material. The fleet composition was observed to be different than the one reported for the Swedish fleet.

**Fleet characteristics for Poland, the Baltic states and Russia**

Local authorities were contacted for existing inventories and surveys for leisure boat activities. Unfortunately, inventories were not available and leisure boat activity in the Baltic Sea for Poland, Lithuania, Latvia, Estonia and Russia were estimated based on marina locations (listed port areas, satellite images) and supporting information. The mix of gasoline and diesel use was taken from the Swedish survey data.

Small harbours along the eastern coast of the Baltic Sea were positioned based on reference locations in http://en.seaclub.lv/ports/estonia/ [link accessed 2018-04] and their size was estimated by satellite images. The total number of crafts for Estonian leisure boats were estimated to be 1075 crafts, which compares well to national registry database (700 yachts + small ships which include 134 motorboats and 206 workboats).

**Appendix B: Fuel consumption estimates for the Swedish fleet and survey data processing**

A large part of the used fleet characteristics originate from the Swedish survey studies (2015), such as the average travel distances and fractions for different engine setups. The survey data is by its nature, qualitative and does not easily provide quantitative information. To overcome this, the survey data has been processed so that it is more usable for the modelling.

The first step in this process was the removal of "blank" answers such as ("Not sure"), which were selected quite often by the users. We simply assumed that the users with blank answers would follow the same distribution as the other users did with

their answers. In other words, we scale the amount of non-blank answers so that they in total sum up to 100 % of all questionnaire users. Such a blank-removed questionnaire summary for fuel consumption habits has been shown in Table B1.

**Table B1: Swedish leisure boat survey material (fuel consumption and boat counts) and derivatives processed based on the data. "L" corresponds to litre.**

|  |  | OSB | MB | LMB | LMSB |
|---|---|---|---|---|---|
| **Boats total (Sweden)** | uses gasoline | 58002 | 181195 | 71164 | 12517 |
|  | uses diesel | 24929 | 17788 | 31556 | 35886 |
| **of which at the Baltic** |  | 29.4 % | 61.6 % | 48.5 % | 70.2 % |
| **Specified gasoline usage** | **Quantitative (L)** | **OSB** | **MB** | **LMB** | **LMSB** |
| **0 – 25 L** | 12.5 | 75.1 % | 32.0 % | 13.2 % | 70.5 % |
| **25.1 – 75 L** | 50 | 19.2 % | 32.0 % | 25.6 % | 27.7 % |
| **75.1 – 250 L** | 162.5 | 5.6 % | 26.4 % | 48.9 % | 1.8 % |
| **250.1 -1000 L** | 625 | 0.1 % | 9.7 % | 9.6 % | 0.0 % |
| **>1000 L** | 1500 | 0.0 % | 0.0 % | 2.9 % | 0.0 % |
| **Boats (Baltic, gasoline)** |  | 24417 | 111631 | 34547 | 8787 |
| **Specified Diesel usage** | **Quantitative (L)** | **OSB** | **MB** | **LMB** | **LMSB** |
| **0 – 25 L** | 12.5 | 100.0 % | 48.0 % | 21.7 % | 42.3 % |
| **25.1 – 75 L** | 50 | 0.0 % | 22.3 % | 26.6 % | 34.7 % |
| **75.1 – 250 L** | 162.5 | 0.0 % | 7.5 % | 20.2 % | 17.2 % |
| **250.1 -1000 L** | 625 | 0.0 % | 10.4 % | 16.3 % | 5.7 % |
| **>1000 L** | 1500 | 0.0 % | 11.9 % | 15.3 % | 0.0 % |
| **Boats (Baltic, Diesel)** |  | 0 | 10959 | 15319 | 25192 |

We transform the qualitative answer possibilities (e.g., 0-25L) into quantitative using the average value of the specified range (e.g., 12.5L). For the last fuel consumption range (>1000L) the averaging is not possible, and we have assumed a value of 1500 liters as the quantitative value. For other statistics such as travel distances we have assumed a 150% value for the last answer option in case the range has been left open in a similar fashion.

Based on the survey material the total amount of gasoline and diesel boats have been specified for each boat class. In addition,
the fraction of boats at the Baltic Sea has also been specified, which in turn yields the amount of gasoline- and diesel powered
boats at the Baltic. Finally, the total fuel consumption estimates for each boat class can be computed by combining a) the boat
counts, b) the quantitative fuel consumption thresholds and c) the distribution of boat owner answers. These totals has been
shown in Fig. B1 and for comparison the BEAM model predictions has been also shown in the figure. For the comparison the
fuel consumption totals have been presented in metric tons (t), which we have obtained by from the quantities in litres by
using a density of 0.8kg/litre for both Diesel and gasoline.

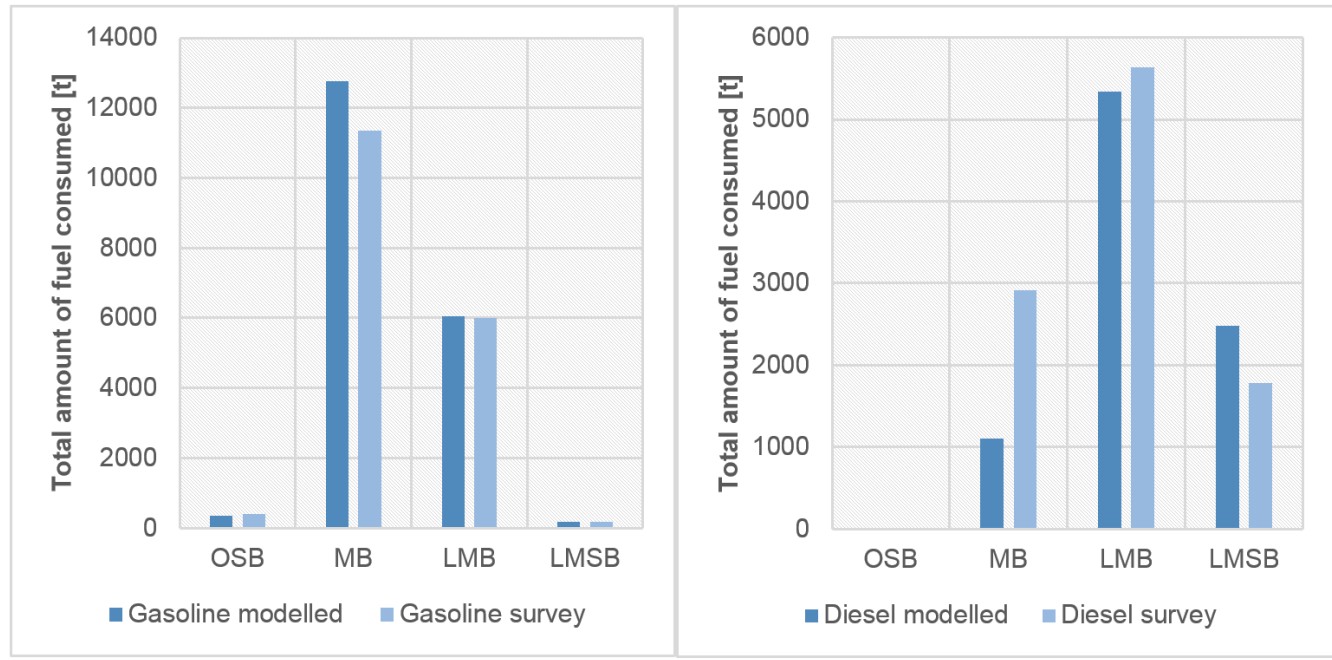

**Figure B1: Comparison of fuel consumption statistics for the Swedish fleet at the Baltic Sea against BEAM model predictions.**
**Comparison has been made for each both class and separately for gasoline and diesel setups.**

**6. Code and data availability**

BEAM model source-code has been written in Java as an extension module under the STEAM model software since they
share common operations and methods to function. STEAM is an intellectual property of the Finnish Meteorological Institute
and it is not freely available due to copyright reasons. The dissemination of input datasets, such as the vessel activity and the
ship technical data, are governed by bilateral contracts with data providers and as such cannot be made available. As such,
the BEAM model is not available as a stand-alone, open source version.
The output data presented in this paper, such as the gridded annual total emissions in netCDF format, are available upon
request from the corresponding author.

**7. Author contribution**

**L. Johansson** designed and carried out the technical BEAM model development described in the paper and processed the
input data required for the shipping emissions modelling. He is also responsible for the shipping emissions modelling work,
results preparation and figures presented in the paper. He prepared the manuscript with contributions from all co-authors. **E.**
**Ytreberg** was responsible for antifouling sections, including the work required to define emission factors for antifouling

paints. He also contributed to manuscript preparation, the Swedish fleet survey material analysis and in the preparation of marina location data sets. **J.-P. Jalkanen** contributed to manuscript preparation and was responsible for the review and analysis of the Finnish leisure boat fleet characteristics. **E. Fridell** contributed to manuscript preparation and in the processing of fleet characteristics that were used in the modelling, especially with regards to emission factors, engine setups and fuel consumption.

**K. M. Eriksson** was responsible for the Swedish marina survey that was conducted as a part of this study for Section 2.4.1. He also contributed to manuscript preparation. **M. Lagerström** contributed to manuscript preparation and antifouling emission factors used in the study. **I. Maljutenko** was responsible for the marina location analysis and fleet composition research for Russia, Estonia, Latvia, Lithuania and Poland. He also contributed in manuscript preparation. **U. Raudsepp** contributed in manuscript preparation. **V. Fischer** was responsible for the fleet characteristics of the German leisure boat fleet based on a survey that was conducted and analyzed as a part of this study. **E. Roth** was responsible for the research on fleet characteristics of the Danish leisure boat fleet.

## 8. Acknowledgements

This work resulted from the BONUS SHEBA project and it was supported by BONUS (Art 185). We are grateful to the Helcom member states for allowing the use of Helcom AIS data in this research.

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
