# Peer review of "Model for leisure boat activities and emissions - implementation for the Baltic Sea"

_Ocean Science, 2020_

## Referee Comment (RC1) · Anonymous Referee #1 · 1 Mar 2020

The publication offers results of a sound and scientif based model, and the results will help to assess the rsiks of shipping anf leisure boats in the baltic Sea

---

## Referee Comment (RC2) · Anonymous Referee #2 · 7 Apr 2020

Dear Editor,

this MS presents a modelling simulation for leisure boat emissions in the Baltic Sea. Addressing leisure boats fills a much needed research gap, when compared to larger vessels. One innovative and very useful contribution, in my view, is the estimation of metal emissions (Cu and Zn) from anti-fouling paints. Proxies are used to validate the model (e.g., AIS-based fuel consumption), which supports the model's robusness. My main concern is whether Zn and Cu emissions from ships at port were included, which is unclear to me when reading the text. Otherwise, the manuscript may be accepted for publication after review.

Specific comments: - Please review the English, for small errors, e.g. * line 32, "utilizes" should be "utilize" * line 33, same for "combines" * line 46, "fail" should be "fails" * line

64, "some studied" should be "some studies"

- line 91, how are the bins defined? Which boat characteristics define a bin? Please clarify

- lines 126-127, "Secondly, the emission factors for contaminants are affected by the geographical distribution of the marina (different paints and release rates are applied)." Please clarify, does this mean that, for example, boats located close to open sea have higher release rates dure to more intense waves, than those located at more protected locations?

- line 127, "specifically on the amount of days spent at sea (ðİŚąðİŚǎ). ", this is not entirely correct, given that metals are released as long as the boat is in contact with water, i.e., also when the ship is at port. Have the authors considered the emissions while at port? This is especially relevant for water quality at the ports, given the lower dispersion of pollutants (and thus, higher concentration rates) than in open sea. Overall, not including release in the port waters will result in an underestimation by the model, which should be highlighted as a potential limitation.

- line 201, what are "otto engines"?

- line 204, "the emission factors for CO and NMHC for older Otto-engines are very high", minor clarification, do the authors mean that the EMEP/EEA emission factors for these engines are much higher than expected for teh Swedish fleet?

- Table 3: antifouling paints are assumed to be used, do the authors have specific information on each boat? I assume that this level of detail was not possible, which is understandable. Please highlight this as a limitation, in addition to the possibility of traces of older paints (anti-fouling and others, and therefore different to the ones in Table 3) still remaining on the hull (although their impact would be minimal).

- Table 4, just to clarify, average values were used in regions where more than one type of paint was expected? For example, in Southern Sweden, the authors used the

average of paints A, B and C?

- lines 341-349, this is a very smart approach, congratulations on having identified this potential source of model underestimation.

- Table 5, please clarify the meaning of the column "Fleet composition type (%)", adding a space after each comma (otherwise it looks like a single number and is quite confusing)

- line 442, "causes the main source of releases to be stationary boats at the marina locations", please clarify, are the AFP emissions from stationary boats also included? Please see my comment above, as the text in line 127 seemed to suggest the opposite.

- section 3.1 (comparison between leisure and commercial shipping): can the authors elaborate on the differences between Zn and Cu emissions, from both types of boats? Are the differences between commercial ships and leisure boats due the surface coated by he pains, or do commercial ships use different types of anti-fouling paints? Is the regulation different for both types of vessels?

- line 510, are these emissions for the year 2016? Or the average per year for the study period?

- line 514, "Also the impact on air quality", are any data available on the release of Cu and Zn to ambient air, during maintenance operation of vessels in the marinas?

---

## Author Comment (AC1) · 8 May 2020

Response to reviewers os-2020-5, Submitted on 16 Jan 2020 Model for leisure boat activities and emissions – implementation for the Baltic Sea

Referee comments 1 The publication offers results of a sound and scientific based model, and the results will help to assess the risks of shipping and leisure boats in the Baltic Sea [edited for readability] We thank the reviewer for this kind assessment!

---

## Author Comment (AC2) · 8 May 2020

Response to reviewers os-2020-5, Submitted on 16 Jan 2020 Model for leisure boat activities and emissions – implementation for the Baltic Sea

==============================================================
NOTE: the response is also provided in a more read-able form as a PDF-attatchment (same content).
==============================================================

Referee comments 2 Dear Editor, this MS presents a modelling simulation for leisure boat emissions in the Baltic Sea. Addressing leisure boats fills a much needed research gap, when compared to larger vessels. One innovative and very useful contribution, in my view, is the estimation of metal emissions (Cu and Zn) from anti-fouling paints. Proxies are used to validate the model (e.g., AIS-based fuel consumption), which supports the model's robustness. My main concern is whether Zn and Cu emissions from ships at port were included, which is unclear to me when reading the text. Otherwise, the manuscript may be accepted for publication after review.

- Thank you! The answer in short is yes, and we have commented on this issue more thoroughly in the additional comments. =========================== Specific comments

*line32,"utilizes" should be "utilize" * line 33, same for "combines" * line 46, "fail" should be "fails" * line, 64, "some studied" should be "some studies"

- The suggested corrections have been now done in the revised manuscript. =========================== line 91, how are the bins defined? Which boat characteristics define a bin? Please clarify

- The bins have been defined as follows: Based on the Swedish survey report there are 4 distinguishable boat classes for which the survey data is presented. To be able to utilize the survey data effectively, we adopted this same boat classification in the model (Table 1). Secondly, each of these boat classes can have up to 5 different engine setups as described in the paper (Table 2). The smallest boat class, the open small boat (SMB) do not use diesel engine setups, however, and it has only 3 possible engine setups. Taking into account the amount of boat classes (4) and all possible engine setups for each class (5) we have 4x5 -2 = 18 different sub classes which we call as "bins" in the model. One of the reason we call them bins here, relates to the technical side of the modelling, where we distribute all boats at marina to these bins so that we can achieve the intended distribution of boat classes and engine setups (Table 2).

-We added a brief clarification to line 91 about the bins, in particular about the characteristics that define the bins. =========================== lines 126-127, "Secondly, the emission factors for contaminants are affected by the geographical distribution of the marina (different paints and release rates are applied)." Please clarify, does this mean that, for example, boats located close to open sea have higher release rates due to more intense waves, than those located at more protected locations?

- With this sentence we simply refer to the difference of emission factors due to salinity and used paint grades that we present in Section 2.3.1 (Antifouling). Technically, the location is the key defining factor for these in the model. We have revised this line (126) to emphasize the paint grade and salinity to make our point clearer. =========================== line 127, "specifically on the amount of days spent at sea (equation). ", this is not entirely correct, given that metals are released as long as the boat is in contact with water, i.e., also when the ship is at port. Have the authors considered the emissions while at port? This is especially relevant for water quality at the ports, given the lower dispersion of pollutants (and thus, higher concentration rates) than in open sea. Overall, not including release in the port waters will result in an underestimation by the model, which should be highlighted as a potential limitation.

- With this commented sentence we refer to the dynamic emission factors that we present in the paper later on. These emission factors are a function of boat-specific counter "days spent at sea", which are presented in Fig 2. We fully agree with the reviewer and we are also confident that the modelling of antifouling paint leech is done in a way that address specifically the "time all boats are in contact with water".

- Considering this and the previous referee comment, we have addressed this comment in the revised paper to make this intention clearer at line 128. The reviewer also raises an important topic related to the dispersion of pollutants and how possibly the dispersion of contaminants should behave differently for port areas and at open sea conditions. However, in this paper we have aimed to present a model capable of estimating the emissions so that perhaps in the future in another paper we could give this
input to dispersion models. We have hinted about this future possibility in the introduction as well as in the conclusions. ============================ line 201, what are "otto engines"?

- We refer to the internal combustion engine that uses the "Otto cycle". Since this detail may be confusing and is in essence redundant, we feel that it is best to remove this unnecessary description (this has been done in the revised manuscript). =========================== line 204, "the emission factors for CO and NMHC for older Otto-engines are very high", minor clarification, do the authors mean that the EMEP/EEA emission factors for these engines are much higher than expected for the Swedish fleet?

- The intention here is to note that the given emission factors (which we use as input) for these above mentioned species and engine types are relatively speaking very high. They will have strong implications for the modelling results and conclusions based on them. For example, from Table 2 one can see that the emission factors for NMVOC's can be more than 5 times higher for "2S" than is shown for the newer "2S 2003". For CO the older 2-stroke gasoline engine has 2 to 3 times larger emission factor. We have addressed this comment in the revised paper to make this intention clearer. Now starting from line (205) we have written: "...the emission factors of Table 2 for 2-stroke gasoline engines for CO and NMVOC are very high; for NMVOC the gasoline engines in general have a significantly larger emission factors than the Diesel engines. Conversely, the older Diesel engines have clearly the highest NOx emission factors." =========================== Table 3: antifouling paints are assumed to be used, do the authors have specific information on each boat? I assume that this level of detail was not possible, which is understandable. Please highlight this as a limitation, in addition to the possibility of traces of older paints (anti-fouling and others, and therefore different to the ones in Table 3) still remaining on the hull (although their impact would be minimal).

- The reviewer is correct, it is not possible to have boat-specific information on used

paint grades for the modelling. Even for commercial shipping geographically de-
fined averaged paint grades need to be used in the modelling since detailed in-
formation on a vessel-level is not available. We have addressed this comment
by elaborating the limitations more thoroughly in the paper (see line 232 – 238).
========================= Table 4, just to clarify, average values were used
in regions where more than one type of paint was expected? For example, in Southern
Sweden, the authors used the average of paints A, B and C?

-Correct! And this is now clarified in the heading of table 4.
========================= lines 341-349, this is a very smart approach,
congratulations on having identified this potential source of model underestimation.

- We thank the reviewer for this kind remark! ========================= Ta-
ble5, please clarify the meaning of the column "Fleet composition type(%)",adding a
space after each comma (otherwise it looks like a single number and is quite confus-
ing)

- We agree and the suggested changes have been made in the revised manuscript. In
the table description we also wrote: ". . .The described fleet composition corresponds to
the percentages used for SB, MB, LMB and LMSB types, in the given order summing
to 100%." ========================= line 442, "causes the main source of
releases to be stationary boats at the marina locations", please clarify, are the AFP
emissions from stationary boats also included? Please see my comment above, as the
text in line 127 seemed to suggest the opposite

- Shortly put, yes they are. As the reviewer points out this question relates to the ear-
lier comments regarding the modelling of berthing boats at marina. We hope that the
revisions done based on the earlier comments are sufficient to address this issue so
that the readers can easily understand the modelling approach that is used also for
berthing boats. ========================= section 3.1 (comparison between
leisure and commercial shipping): can the authors elaborate on the differences between Zn and Cu emissions, from both types of boats? Are the differences between commercial ships and leisure boats due the surface coated by the paints, or do commercial ships use different types of anti-fouling paints? Is the regulation different for both types of vessels?

- The higher loads of copper and zinc from the commercial fleet can primarily be explained by legislation and use pattern, which is now clarified in the manuscript (line 487 – 491). ========================= line 510, are these emissions for the year 2016? Or the average per year for the study period?

- For these comparisons against commercial shipping we have used AIS-data for 2014 and the STEAM model. We have elaborated this in the revised manuscript (lines 516-518).

-Our modelling results (concentrating on air emissions) for the Baltic Sea are publicly available via Helcom (See HELCOM Maritime19/5-2.INF at available at: https://portal.helcom.fi/meetings/MARITIME%2019-2019-582/MeetingDocuments/5-2%20Emissions%20from%20Baltic%20Sea%20shipping%20in%202006%20-%202018.pdf).

- It should be mentioned that the commercial fleet emissions are fairly similar on an annual level across the Baltic for 2012-2018. Therefore, the main conclusions would be the same regardless of the year that is chosen for this comparison. ========================= line 514, "Also the impact on air quality", are any data available on the release of Cu and Zn to ambient air, during maintenance operation of vessels in the marinas?

- In this paper for leisure boat emissions we have two views: one for exhaust emissions (NOx, PM2.5, NMVOC, CO) and one for water emissions (Zn and Cu). In this particular line we point out that the modelled exhaust emissions especially for NMVOC are quite substantial for the summer months. This is something that "should be studied further with measurements and dispersion modeling". To our knowledge such data is

not available for the release of Cu and Zn to ambient air and we suspect that these pollutants are not transferred to atmosphere with any meaningful quantities from this type of a source. ===========================

---

## Author Response (AR2)

**Response to Editor Comments**

Topic Editor Decision: Publish subject to technical corrections (22 Jul 2020) by David Turner

Comments to the Author:

This is a pioneering work. The authors have addressed the Reviewers' comments satisfactorily. Some small technical corrections should be made:

1. **Practical salinity is a dimensionless quantity, so the "unit" PSU should be deleted throughout the manuscript**

We have removed most of the PSU units in the paper. In line 225 in which this "unit" is introduced the first time we write "…in Gothenburg (salinity, PSU, 14)" so that the reader knows we refer to practical salinity unit in the paper. We also mention "PSU" in the header of Table 4 once to make it clearer.

2. **The unit "ton" (imperial ton = 2240 lb) is used through most of the manuscript. I am aware that old habits die hard in the shipping industry and that the use of this unit may well persist. A note at the start of the paper would help to clarify that this old unit is used rather than "tonne" (= 1000 kg).**

Thank you for this comment. We have clarified in the text that we do refer to "tonne" throughout the text. For example in Table 6 we changed the units to [10^3 kg]. In the text below the table we use the unit "t". We also removed the unit "kton" that was previously used in the text.

3. **On a point of consistency, the adjective "riparian" is sometime capitalised, sometimes not. Consistent use is important: the uncapitalised version is to be preferred.**

Thank you, changes have been made as suggested.

4. **Lines 115 and 122: better to avoid the use of "species", which also has biological connotations. "contaminant", also used later in the text, is a good alternative**

The word "species" have been replaced with better alternatives, as suggested.

5. **Line 128: using the term "days spent at sea" does not fully address the Reviewer's concern since the implication for many readers is the number of days the boat is being actively used. "days spent in the water" would be much clearer**

Thank you, this change does make it clearer.

6. **Lines 190-191: the load factor (70%) is the same as quoted on line 183, i.e. not higher**

We added slight clarifications to these lines. In Table 1 we present our values in which we base our computations. The survey data for OSB average engine load did not seem consistent with the other statistics and therefore we adjusted it upwards to 70% from 50%. We hope this is now clearer.

**7. Line 335: should the second date be 3rd June?**

According to our, admittedly, very limited statistics we have four dates: (1) the start date, (2) the day when 100% capacity is reached, (3) the day when the capacity starts to reduce gradually and (4) the date when capacity falls to 0. In this line 335 we refer to this "inner" date of (3) which is not in early June.

**8. Table 6: I assume that the final column is millions of km, "10^6 km" would be clearer**

Indeed, thank you!